# VSSFlow: Unifying Video-conditioned Sound and Speech Generation via Joint Learning

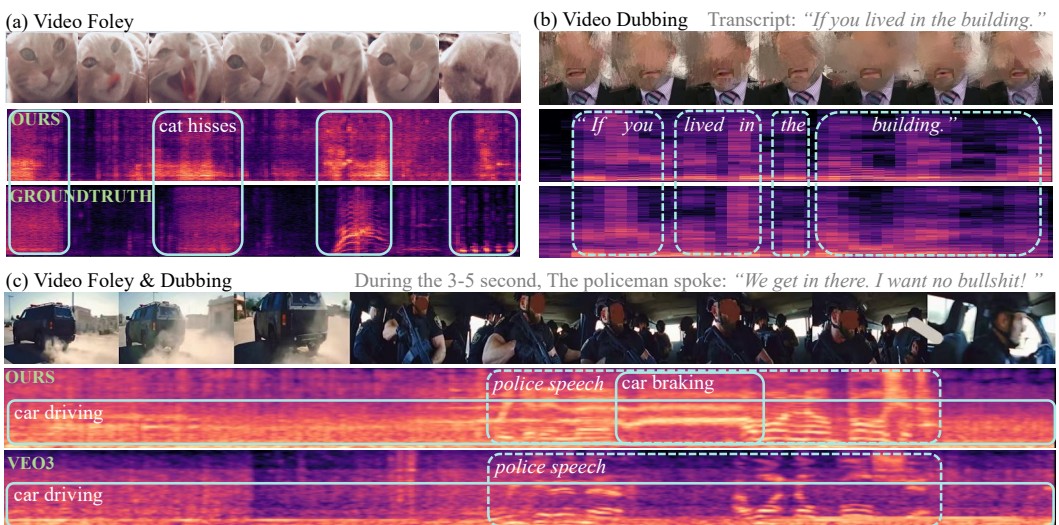

Figure 1: VSSFlow: A unified generative model for video-conditioned sound and speech synthesis. (a) Video-to-Sound generation given a silent video. (b) Visual-TTS generation given a silent talking video and speech transcripts. (c) Sound-Speech Joint generation given a silent video and transcripts.

## Abstract

Video-conditioned sound and speech generation, encompassing video-to-sound (V2S) and visual text-to-speech (VisualTTS) tasks, are conventionally addressed as separate tasks, with limited exploration to unify them within a signle framework. Recent attempts to unify V2S and VisualTTS face challenges in handling distinct condition types (e.g., heterogeneous video and transcript conditions) and require complex training stages. Unifying these two tasks remains an open problem. To bridge this gap, we present **VSSFLOW**, which seamlessly integrates both **V2S** and **VISUALTTS** tasks into a unified **FLOW**-matching framework. VSSFlow uses a novel condition aggregation mechanism to handle distinct input signals. We find that cross-attention and self-attention layer exhibit different inductive biases in the process of introducing condition. Therefore, VSSFlow leverages these inductive biases to effectively handle different representations: cross-attention for ambiguous video conditions and self-attention for more deterministic speech transcripts. Furthermore, contrary to the prevailing belief that joint training on the two tasks requires complex training strategies and may degrade performance, we find that VSSFlow benefits from the end-to-end joint learning process for sound and speech generation without extra designs on training stages. Detailed analysis attributes it to the learned general audio prior shared between tasks, which accelerates convergence, enhances conditional generation, and stabilizes the classifier-free guidance process. Extensive experiments demonstrate that VSSFlow surpasses the state-of-the-art domain-specific baselines on both V2S and VisualTTS benchmarks, underscoring the critical potential of unified generative models.

**Project Page:** `https://vasflow1.github.io/vasflow/`

## 1 INTRODUCTION

Multimodal generative models have achieved remarkable progress recently, with sound (Iashin & Rahtu, 2021), speech (Chen et al., 2022), and video (Wan et al., 2025; Guan et al., 2025) generation emerging as central components of multimedia content creation. In industry, multimodal systems are now capable of generating sound and speech jointly from the input video (DeepMind, 2024). Veo3 (Google, 2025) even shows remarkable performance of sound, speech, and video joint generation capability. In contrast, academic research mainly focuses on the single task: video-to-sound (V2S [1]), which generates synchronized and semantically-aligned sound but struggles to produce intelligible speech, and visual text-to-speech (VisualTTS), which generates high-quality lip-sync speech but is constrained in portrait-style videos and fail to generate non-speech sound. This work takes a step further to address the missing link between video-conditioned sound and speech generation under one unified framework, supporting both V2S and VisualTTS tasks, as well as their combination, i.e, generating speech alongside environmental sound simultaneously.

Recent efforts have aimed to unify both V2S and VisualTTS tasks. AudioGen-Omni (Wang et al., 2025) employs a single flow-based model to process all video and text conditions through in-context conditioning (concatenating or adding conditions to noisy audio tokens). However, we find this approach suboptimal. Using the same in-context condition mechanism for features from different modalities fails to capture the distinct interactions between video, text, and audio. DualDub (Tian et al., 2025), an autoregressive model, treats text as a prefix and combines video conditions with audio tokens via an additional fusion module. While this design effectively handles multiple conditions, it introduces a complex training procedure that relies on curriculum learning to progressively acquire speech and sound knowledge. Consequently, unifying V2S and VisualTTS remains an open challenge, particularly in managing diverse condition types and efficiently learning both sound and speech knowledge within a single model.

To address these challenges, we introduce VSSFlow, a flow-based framework (Lipman et al., 2022) that unifies V2S and VisualTTS generation. Firstly, to handle both video and transcript conditions, we systematically explore the optimal conditioning mechanism within the Diffusion Transformer (DiT) (Peebles & Xie, 2022) block. The two types of conditions exhibit inherently different characteristics: speech transcripts provide more deterministic guidance for speech generation, whereas video features are more ambiguous for sound generation. Also, two typical types of conditioning mechanisms lead to distinct inductive biases, i.e., cross-attention conditioning processes ambiguous video features well, while in-context/self-attention conditioning better exploits transcript features. Accordingly, VSSFlow integrates transcript embeddings by concatenating with audio latents and processing them through self-attention, while incorporating video representations via cross-attention layers. Secondly, under our VSSFlow framework, we observe that no complex design for training stages is necessary. The joint learning of sound and speech does not lead to interference or performance degradation, a common issue in multitask learning (Kendall et al., 2018; Tian et al., 2025). Instead, it results in mutual enhancement on both tasks. We conduct analysis and reveal that this improvement stems from the model's ability to learn general audio knowledge, which leads to faster convergence, better conditioned generation results, and more stable classifier-free guidance process.

Beyond unifying V2S and VisualTTS tasks, real-world scenarios often require generating sound and speech jointly. We further apply continual training on the VSSFlow model with synthetic sound-speech mixed data (i.e., speech with environmental sounds). As shown in Fig 1(c), the model rapidly adapts to this setting, successfully generating mixed outputs for out-of-domain videos —e.g., generating car engine sounds alongside a police man speaking clearly.

Evaluations on V2S and VisualTTS benchmarks show that VSSFlow achieves exceptional sound fidelity and speech quality. VSSFlow lays a powerful foundation for video-conditioned sound and speech generation. In summary, our contributions are as follows:

(1) We introduce VSSFlow, a unified framework for video-conditioned sound and speech generation, with an effective condition aggregation mechanism for integrating video and speech features into DiT blocks and handling both V2S and VisualTTS tasks. (2) Through systematic analysis, we

---

[1]In prior works, the term "Video-to-Audio (V2A)" has commonly been used. To avoid ambiguity in this paper, we define "sound" as non-linguistic audio, such as environmental or natural audio, distinct from speech. "Speech" refers to audio containing linguistic information, such as spoken language. "Audio" is used to encompass sound, speech, and all other types of auditory content.

demonstrate that under the flow model's end-to-end training setting, the joint learning of sound and speech generation produces a mutual promotion effect, highlighting the critical role of unified models in the sound and speech generation field. (3) Extensive evaluations confirm VSSFlow's superior performance, surpassing the SOTA domain-specific baselines on V2S and VisualTTS benchmarks.

## 2 RELATED WORK

### 2.1 VIDEO-TO-SOUND AND VISUAL TEXT-TO-SPEECH GENERATION

Video-to-Sound (V2S) generation aims to produce environmental or natural sound, which is semantically- and temporally-aligned with the given silent video. Recent advances in generative models have driven significant progress in the V2S field, with approaches categorized into three main paradigms: Autoregressive models (Sheffer & Adi, 2022; Iashin & Rahtu, 2021; Mei et al., 2024; Viertola et al., 2025) convert sound into discrete tokens and predict the sound token sequence conditioned on video features; Mask-based methods (Liu et al., 2024b; Pascual et al., 2024; Su et al., 2024b) also treat sound as tokens and reconstruct the sequences by predicting masked tokens; Flow- or diffusion-based methods (Luo et al., 2023; Zhang et al., 2024; Wang et al., 2024b; Cheng et al., 2024; Wang et al., 2024c; Xing et al., 2024; Cheng et al., 2025) iteratively transform noise into a waveform guided by video conditions, achieving high generation fidelity. Another related task, visual text-to-speech (VisualTTS) generation, aims to generate speech that is consistent with the given video in aspects like speaker's style, lip movements, emotions, and so on. These models (Chen et al., 2022; Cong et al., 2023; 2025; Lu et al., 2023; Cong et al., 2024b; Hu et al., 2021) are often compact in size, and built on mature TTS frameworks like FastSpeech2 (Chien et al., 2021) and MatchaTTS (Mehta et al., 2024). By incorporating multiple prosodic features like pitch, energy, emotional cues and so on, these models demonstrate powerful speech generation capability.

Speech generation focuses on capturing speaker characteristics and ensuring linguistic accuracy, while sound generation aims to accurately reproduce various non-speech audio types. Consequently, these two related fields are typically treated as distinct domains. Most V2S models fail to produce intelligible speech, and VisualTTS models are unable to generate non-speech sounds. In this paper, we propose a unified framework to address both tasks, aiming to develop a model that efficiently introduces different conditional signals and performs comparably to domain-specific baselines.

### 2.2 UNIFIED MODELS FOR VIDEO-CONDITIONED SOUND AND SPEECH GENERATION

Some advancements have been made to integrate video-conditioned sound and speech generation into unified models. For example, Audiobox (Vyas et al., 2023) from Meta and V2S model (Deep-Mind, 2024) from Google leverage diffusion models to generate high-fidelity sound and speech from multimodal prompts. More recently, Veo3 (Google, 2025) has garnered significant attention for its ability to generate videos with synchronized background sound and human speech, sparking renewed interest in unified visual-sound-speech generation models. In the academic community, AudioGen-Omni (Wang et al., 2025) integrates multiple video and text prompts all through in-context conditioning into a unified flow model. DeepAudio (Zhang et al., 2025) and DualDub (Tian et al., 2025) have explored unified video-conditioned audio generation by employing fusion modules, such as large language models (LLMs), to integrate speech and sound generation heads. They rely on multi-stage training strategies, where the model is progressively trained to acquire distinct generation capabilities for speech and sound.

The interplay between sound and speech generation poses a critical question for unified audio generation, yet it remains underexplored in prior work. Are sound and speech, as distinct audio modalities, entirely separate tasks, or can they be effectively learned jointly? A common belief is that end-to-end joint training degrades generation performance compared to curriculum learning approaches (Tian et al., 2025). In contrast, our proposed VSSFlow framework offers a novel perspective: end-to-end joint learning of sound and speech within a flow-matching framework does not lead to mutual suppression. Instead, it fosters a synergistic effect, driven by the shared knowledge between the sound and speech modalities. By leveraging a tailored condition aggregation mechanism and end-to-end joint learning, VSSFlow effectively balances the distinct requirements of V2S and VisualTTS tasks, achieving superior performance across diverse video contexts.

## 3 METHOD

### 3.1 PRELIMINARY

In this section, we provide an overview of the flow-matching model and classifier-free guidance, which serve as the foundation for our proposed VSSFlow framework.

**Flow-Matching Framework.** Flow-matching (Lipman et al., 2022) transforms a source distribution $\mathcal{P}(x_0)$, typically Gaussian noise $x_0 \sim \mathcal{N}(0, 1)$, into a target audio distribution $\mathcal{P}(x_1)$. This process is governed by a continuous-time ODE with learned velocity field $v_\theta(x_t, c, t)$, where $t \in [0, 1]$, $x_t$ is the sample state at timestep $t$, $c$ is an optional condition, and $\theta$ denotes parameters:

$$\frac{dx_t}{dt} = v_\theta(x_t, c, t), \quad x_t = x_0 + \int_0^t v_\theta(x_s, c, s) \, ds. \tag{1}$$

In our unified video-conditioned sound and speech generation setting, condition $c$ mainly includes video representations $c_v$ and speech transcript representations $c_p$. The training objective minimizes the difference between predicted and ground-truth velocities along the optimal transport path:

$$\mathcal{L}_{\text{FM}} = \mathbb{E}_{t,x_0,x_1} \|v_\theta(x_t, c, t) - \dot{x}_t\|^2, \tag{2}$$

where $x_t = tx_0 + (1 - t)x_1$ and $\dot{x}_t = x_1 - x_0$. Flow-matching enables efficient sampling via ODE solvers, enhancing convergence and computational efficiency.

**Classifier-Free Guidance.** In inference, Gaussian noise $x_0 \sim \mathcal{N}(0, 1)$ is sampled, and the model uses condition $c$ to progressively refine $x_0$ into a clean latent representation $x_1'$. Classifier-free guidance (CFG) (Ho & Salimans, 2022) enhances generation quality in conditional flow-matching models. During training, the model learns both conditional and unconditional modeling ability by randomly setting $c = \emptyset$. In inference, predictions at each timestep $t$ are interpolated as:

$$v_\theta^{\text{CFG}}(x_t, c, t) = v_\theta(x_t, \emptyset, t) + \gamma(v_\theta(x_t, c, t) - v_\theta(x_t, \emptyset, t)), \tag{3}$$

where $\gamma$ is the guidance scale controlling the strength of the condition.

### 3.2 VSSFLOW FRAMEWORK

In this section, we introduce VSSFlow's framework and different mechanisms within its DiT block to introduce multiple conditions, including video and speech transcript representations.

**Model Overview.** As illustrated in Figure 2, VSSFlow adopts 10-layer cross-attention-based DiT for unified video-conditioned sound and speech generation. Each DiT block follows the implementation in Stable Audio Open (Evans et al., 2025). Besides, to better capture temporal relationships between the sound sequence and video frame sequence in the V2S process, we incorporate 1D Rotary Position Embedding (RoPE) (Su et al., 2024a) into the query and key matrix in both self- and cross-attention blocks. The waveform is converted to a melspectrogram and processed by the VAE encoder to get $x_0$. The denoised latent $x_1'$ is decoded to a melspectrogram by the VAE decoder and then converted into a waveform via a vocoder. More details can be found in Appendix B.

**Condition Representation for Video and Speech.** Video-conditioned unified audio generation relies on two primary condition signals: video frame representations and speech transcripts. These signals differ fundamentally in their relationship to the generated audio. Speech transcripts exhibit high determinism, as they directly encode linguistic content, resulting in speech outputs that are relatively consistent, with variations mainly in prosody, timbre, and style. Conversely, video frames introduce greater uncertainty, as identical visual inputs can map to a diverse set of plausible sound outputs, reflecting the more variable nature of non-linguistic audio. Consequently, V2S generation poses greater modeling complexity compared to the more deterministic VisualTTS process, while VisualTTS generation has higher requirements for the accuracy of the generated speech. These distinct characteristics present challenges for the design of the model's condition mechanism.

In our work, video is encoded by CLIP model (Radford et al., 2021) at 10 FPS, resulting in visual representations $c_v \in \mathbb{R}^{T_v \times D_v}$, where $T_v$ is the frame number and $D_v$ is CLIP embedding dimension.

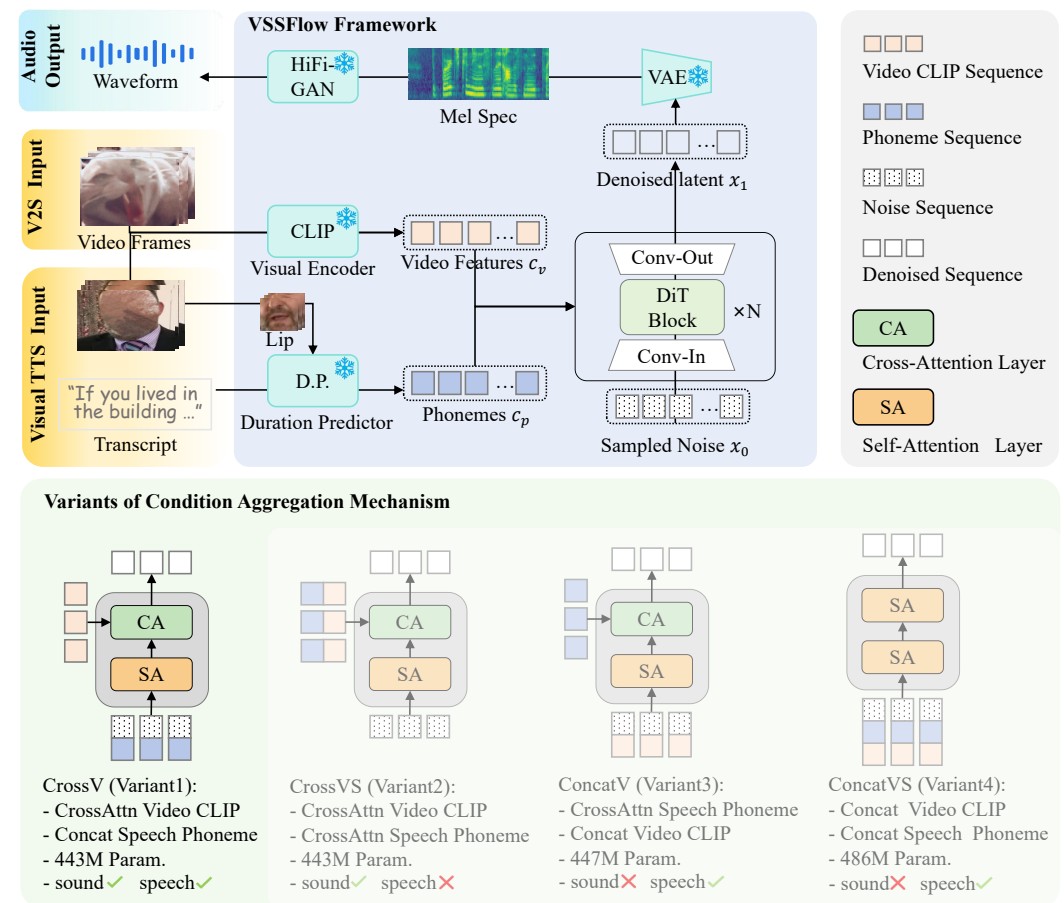

Figure 2: Overview of VSSFlow's architecture. VSSFlow employs cross-attention-based DiT blocks and a flow-matching paradigm, taking video CLIP representations and speech phoneme embeddings as conditional inputs. We conduct ablation studies on different condition mechanisms of DiT illustrated in section 4.2. Variant CrossV (introducing video condition via cross-attention and speech condition via concatenation) enhances overall performance on both V2S and VisualTTS tasks.

Speech transcripts are converted to phoneme sequences representations $c_p \in \mathbb{R}^{T_p \times D_p}$, where $T_p$ is phoneme token sequence length and $D_p$ denotes the phoneme embedding dimension. More details on condition representation can be found in Appendix B.

**Condition Aggregation Mechanism.** In this section, we present the design choices of the condition aggregation mechanism for effectively integrating video and speech representations. Within the cross-attention-based DiT architecture, complex sequence conditions are incorporated through two primary mechanisms: 1) via cross-attention layers, where conditions are used as key and value matrices to guide the generation process, and 2) via concatenation with the initial latent representation, which is processed primarily through self-attention blocks. Cross- and self-attention block has no structural difference, but the self-attention block additionally models the relationships among latent token sequences. As discussed before, to determine the optimal way of introducing different signals, we propose and evaluate four variants of the condition mechanism, as illustrated in Figure 2.

- CrossV (Variant 1): Video representations are introduced in cross-attention layer, and speech representation is concatenated with the latent sequence.
- CrossVS (Variant 2): Both video and speech representations are introduced in cross-attention layer.
- ConcatV (Variant 3): Speech representations are introduced in cross-attention layer, and the video representation is concatenated with the latent sequence.
- ConcatVS (Variant 4): Video and speech representations are concatenated with the latent sequence.

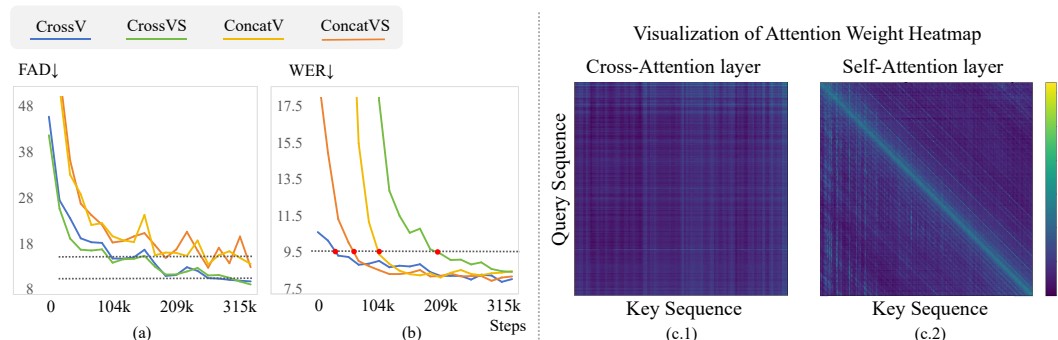

Figure 3: Performance comparison of different conditioning mechanisms over training steps. (a) shows FAD metric for V2S task, while (b) presents WER metrics for the VisualTTS task. (c) is the visualization of the attention weights in self- and cross-attention layers of DiT blocks. More metrics can be found in Appendix C.1.

## 4 EXPERIMENTS

### 4.1 EXPERIMENTS SETUP

In the experiment section, we respond to the unaddressed challenges mentioned above: (1) How to build a unified end-to-end framework for video-conditioned sound and speech generation, and find the optimal method to integrate different types of conditions? (2) Does joint training of speech and sound affect each other's generation performance, and is this impact positive or negative? VSSFlow is trained end-to-end on multiple sound and speech datasets using the flow loss function defined in Equation 2. The data covers three tasks: (1) Video-to-sound (V2S), with phoneme conditions $c_p = 0$; (2) Visual test-to-speech (VisualTTS), with both $c_p$ and $c_v$ are active; And (3) text-to-speech (TTS), with $c_v = 0$. See Appendix D for more details about the dataset and training settings.

### 4.2 COMPARISON OF CONDITION AGGREGATION MECHANISMS

We first explore optimal mechanisms for integrating video and speech conditions, training four variants (illustrated in Section 3.2) for 350k steps (200 epochs) and evaluating them on both V2S and VisualTTS tasks. The main evaluation results are shown in Figure 3, and additional metrics can be found in Appendix C.1. Based on the results, we draw the following key findings:

- Self-attention layers show an inductive bias favoring current and nearby positions, as visualized in attention maps (Figure 3(c.2)). Cross-attention layers (Figure 3(c.1)) lack this localization, reflecting distinct interaction dynamics.

- Aligned phoneme embeddings—which are deterministic and content-tied—suit concatenation-based integration. As shown in Figure 3(b), phoneme-concatenation variants (CrossV, ConcatVS) converge faster. Notably, Variant CrossV reaches very low WER in only 80k steps. As the strong alignment between the speech features and the speech waveform, the concatenation method leverages self-attention's inductive bias to speed convergence and boost speech generation quality.

- Video CLIP features—which are weakly linked to mel-spectrograms—favor flexible cross-attention integration under the setting of sound-video joint training. Figure 3(a) shows that cross-attention of video features (CrossV, CrossVS) outperforms the concatenation methods (ConcatV, ConcatVS). The video features are less determinant for the sound waveform. Therefore, using concatenation to introduce the video features harms the learning process and sound quality due to the aforementioned inductive bias of the self-attention block. Through cross-attention, the model can learn the relationship in a flexible way and enhance sound generation performance.

CrossV variant works as final condition mechanism as better performance on both generation tasks.

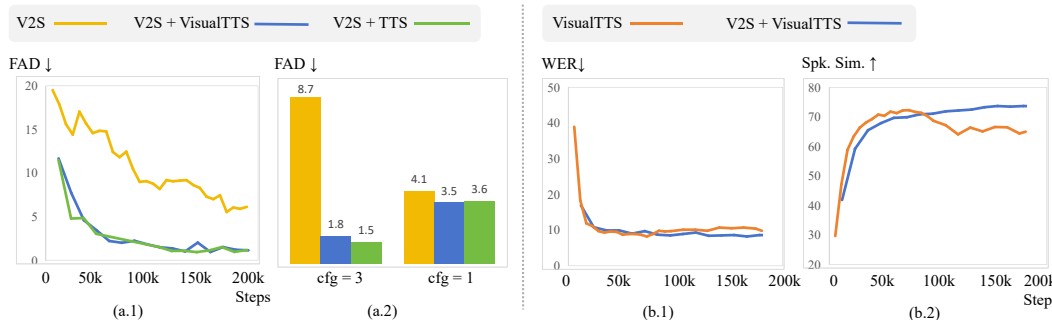

Figure 4: Impact of joint learning on the performance of sound and speech generation. The left three models are trained with different data settings: V2S only, V2S + VisualTTS, and V2S + TTS. (a.1) shows the FAD metric for the sound generation task across training steps. (a.2) compares the performance of different models (trained on three data settings for 100k steps) under varying classifier-free guidance scales. The right two models are trained with VisualTTS data only and V2S + VisualTTS data. (b) plots the WER and Speaker Similarity metrics for the VisualTTS task across training steps. More metrics can be found in Appendix C.2.

### 4.3 EVALUATIONS ON JOINT LEARNING OF SOUND AND SPEECH GENERATION

We further investigate the impact of combining sound and speech data during training. We train VSSFlow under various data configurations, with primary results shown in Figure 4. Additional metrics are shown in Appendix C.2.

**Impact of Speech Data on V2S Performance.** For V2S, we compare three settings: (1) V2S data only, (2) V2S + VisualTTS data, and (3) V2S + TTS data. Results in Figure 4(a.1) show that incorporating speech data — whether visual modality is included or not (VisualTTS or TTS) — enhances sound generation performance. Models trained with both speech and sound data significantly outperform the V2S-only baseline at equivalent training steps.

**Impact of Sound Data on VisualTTS Performance.** For VisualTTS, we train VSSFlow with two data settings: (1) VisualTTS data only and (2) V2S + VisualTTS data. Figure 4(b) shows that the two variants have comparable speech generation performance at the same training step, indicating that joint training with sound does not impair speech generation performance or convergence speed. Moreover, the model with extra sound training data achieves better convergence performance after the same number of epochs.

**Findings.** Contrary to prior work (Tian et al., 2025), our experiments show that joint learning of sound and speech generation is mutually beneficial, not suppressive, within the end-to-end VSS-Flow framework. Including multiple data types boosts performance on both tasks, highlighting the potential of unified generative models for multimodal sound and speech synthesis tasks.

Training on diverse audio types (including both sound and speech) enables VSSFlow to fit the audio distribution more comprehensively, therefore enhancing the efficacy of CFG process. Capturing a more general audio distribution through joint sound-speech learning, the difference between conditional and unconditional prediction produces a more informative direction, improving the guidance quality of CFG. As shown in Figure 4(a.2), we compare models trained for 100,000 steps across three data settings, evaluating their sound generation performance under varying cfg scales. According to Equation 3, at cfg = 1, when the unconditional generation result has not yet been utilized, all models exhibit similar conditional generation capabilities, with joint-training settings perform a little better (3.5, 3.6 vs. 4.1). However, at higher cfg (cfg = 3), when unconditional result is also taken into account, models trained with additional speech data show substantial gains, while the V2S-only model improves minimally or even exhibits degradation. Prior work (Fan et al., 2025) shows that implementing CFG on underfitted models can degrade the generation performance. The narrow distribution of just sound data may contribute to this. However, joint sound-speech training mitigates this phenomenon, highlighting the importance of the shared knowledge across domains.

Table 1: V2S evaluation results on the VGGSound benchmark. "Param.Count" denotes the number of parameters of the generation backbone. "Visual Rep." indicates the type of visual representation for condition – O.F. refers to optical flow, S.AV. refers to Segment AVCLIP, I.B. refers to ImageBind features, and S.F. refers to SynchFormer-extracted features. "Extra Data" lists additional training datasets beyond VGGSound. A.S refers to another commonly used V2S dataset, AudioSet. Except for this, MMAudio includes additional higher-quality text-to-sound data, AudioCaps and WavCaps (A.&W.C.), for training. V2A-Mapper (V2A-M.) and Foleycrafter (FoleyC.) utilize a frozen pre-trained text-to-sound module. Therefore, we label them in gray and do not directly compare with them. For each metric, the highest score is in **bold** and the second-highest score is underlined.

| | Model Information | | | Sound Quality | | | | | Synchronization | | | Seman. |
|---|---|---|---|---|---|---|---|---|---|---|---|---|
| Method | Param. Count | Visual Rep. | Extra Data | FAD ↓ (vgg.) | FAD ↓ (pann.) | FAD ↓ (pas.) | IS ↑ (pas.) | KL ↓ (pas.) | Onset Acc. ↑ | Onset Ap. ↑ | DeSync ↓ | IB-VA ↑ |
| SpecVQ. | 377M | RGB, O.F. | – | 4.82 | 30.35 | 288.98 | 4.12 | 3.21 | 5.59 | 20.77 | 1.23 | 14.26 |
| Im2Wav | 360M | CLIP | – | 4.95 | 20.19 | 258.43 | 5.89 | 2.24 | 6.19 | 19.81 | 1.22 | 19.59 |
| V-AURA | 893M | S.AV. | – | 4.01 | 31.76 | 328.36 | 8.50 | 2.80 | 6.50 | 22.59 | 1.14 | 24.00 |
| VAB | 403M | CLIP | A.S. | 3.05 | 19.72 | 238.81 | 7.51 | 2.37 | 6.31 | 18.45 | 1.18 | 25.67 |
| Difffoley | 859M | CAVP | A.S. | 4.72 | 25.97 | 414.98 | 9.56 | 2.88 | 7.07 | 13.94 | **0.99** | 10.35 |
| Seeing. | 416M | I.B. | – | 3.93 | 22.95 | 245.01 | 5.68 | 2.71 | 6.56 | 16.22 | 1.20 | – |
| V2A-M. | – | CLIP | – | 1.45 | 8.33 | 166.55 | 10.33 | 2.52 | 7.82 | 16.75 | 1.23 | 21.91 |
| FoleyC. | 1126M | CLIP | – | 2.17 | 17.94 | 150.58 | 9.82 | 2.30 | 5.93 | 21.42 | 1.21 | 27.55 |
| TiVA | 346M | CLIP | A.S. | 3.90 | 25.24 | 323.77 | 4.61 | 2.77 | 4.84 | 22.32 | 1.15 | 16.46 |
| LoVA | 1057M | CLIP | A.S. | 2.03 | 17.67 | **120.32** | 9.91 | 2.15 | 7.02 | 19.95 | 1.22 | 26.01 |
| Frieren | 421M | CAVP | – | 2.09 | 29.13 | 214.59 | **12.95** | 2.92 | **7.61** | 18.5 | 1.11 | 22.82 |
| MMAudio | 621M | CLIP, S.F. | A.&W.C. | 1.21 | 10.02 | 153.93 | 11.52 | 2.05 | 7.21 | 24.12 | 0.45 | 32.54 |
| VSSFlow | 443M | CLIP | – | **1.34** | **11.10** | 187.40 | 7.09 | **2.13** | 7.16 | **22.68** | 1.18 | **26.01** |

Table 2: VisualTTS evaluation results on Chem and GRID benchmark. For each metric, the highest score is in **bold** and the second-highest score is underlined.

| Bench | Method | WER ↓ | Spk. Sim. ↑ | UTMOS ↑ | MCD ↓ | MCD-D. ↓ | MCD-DS. ↓ | LSE-C ↑ | LSE-D ↓ |
|---|---|---|---|---|---|---|---|---|---|
| | GT | 3.5 | 100 | 4.19 | 0 | 0 | 0 | 7.66 | 6.88 |
| | GT-vocoder | 3.5 | 93.3 | 3.19 | 3.33 | 2.67 | 2.67 | 7.57 | 6.9 |
| Chem | E2-TTS | 8.6 | 67.4 | 3.43 | 14.73 | 5.89 | 7.21 | 1.5 | 13.17 |
| | E2-TTS-tuned | 8.7 | 70.2 | 3.51 | 14.46 | 5.07 | 6.21 | 1.62 | 13.1 |
| | DSU | 37.3 | 72.9 | 3.33 | 10.52 | 6.7 | 6.73 | 5.88 | 7.86 |
| | HPMDubbing | 22.3 | 44.6 | 3.11 | 11.54 | 5.19 | 6.30 | **7.30** | **7.58** |
| | StyleDubber | 16.3 | 73.7 | 3.14 | 14.46 | 6.01 | 6.36 | 3.58 | 11.08 |
| | EmoDubber | 16.7 | 78.1 | **3.87** | 14.87 | 5.8 | 7.16 | 5.48 | 8.27 |
| | VSSFlow | **15.1** | **79.7** | 3.17 | **9.55** | **5.18** | **5.19** | 6.1 | 8.37 |
| | GT | 12.9 | 100 | 4.04 | 0 | 0 | 0 | 5.49 | 8.56 |
| | GT-vocoder | 13.4 | 64.1 | 3.37 | 5.02 | 3.88 | 3.89 | 6.83 | 8.16 |
| GRID | E2-TTS | 21.4 | 43.2 | 3.57 | 15.73 | 5.62 | 7.05 | 1.93 | 12.29 |
| | E2-TTS-tuned | 20.2 | 44.2 | 3.62 | 15.64 | 5.59 | 6.98 | 2.36 | 12.08 |
| | DSU | 34.3 | 5.9 | 3.55 | 13.82 | 10.55 | 10.57 | 5.63 | 8.73 |
| | HPMDubbing | 27.6 | 31.3 | 2.11 | 12.31 | 8.05 | 8.23 | 6.02 | 8.85 |
| | StyleDubber | **10.9** | 51.4 | 3.74 | 12.85 | 7.81 | 7.91 | 6.33 | 8.77 |
| | EmoDubber | 15.9 | 50.5 | **3.98** | 15.52 | 5.89 | 9.83 | 3.48 | 10.35 |
| | VSSFlow | 18.2 | **51.5** | 3.31 | **8.66** | 5.23 | 5.23 | 6.37 | 8.6 |

## 4.4 BENCHMARK RESULTS

Based on the findings in section 4.2 and 4.3, the reported VSSFlow model is based on architecture CrossV, trained for 260k steps (150 epochs) using a total of 503k data from V2S, TTS, and VisualTTS datasets. For the V2S generation, we evaluate the model's performance on the standard VGGSound benchmark. For VisualTTS generation, we use widely adopted Chem and GRID benchmarks.

**Baselines and Metrics.** For V2S evaluation, we compare VSSFlow against baselines from autoregressive, mask-based, diffusion, and flow-based paradigms as shown in Table 1. Our primary baselines are **Frieren** (Wang et al., 2024c), which has comparable parameters, flow matching paradigm, and uses the same V2S dataset. And **LoVA** (Cheng et al., 2024), which uses more parameters, diffusion paradigm, and is fully initialized from Stable Audio Open. For sound quality, we use Fréchet Audio Distance (FAD) (Kilgour et al., 2018), Inception Score (IS) (Salimans et al., 2016), and Mean KL-Divergence (KL). For temporal alignment, we report wildly-adopted onset accuracy (Onset Acc.) and onset average precision (Onset AP) following prior works (Zhang et al., 2024; Du et al., 2023), plus DeSync Score following (Cheng et al., 2025). See Appendix E for details.

For VisualTTS evaluation, we compare VSSFlow against four SOTA baselines: DSU (Lu et al., 2023), HPMDubbing (Cong et al., 2023), StyleDubber (Cong et al., 2024b), and EmoDubber (Cong et al., 2024a). For speech quality, we use Word Error Rate (WER) and UTokyo-SaruLab Mean Opinion Score (UTMOS). For speech alignment, we compute MCD, MCD-DTW, and MCD-DTW-SL. For speech-visual alignment, we report Lip Sync Error Distance (LSE-D) and Lip Sync Error Confidence (LSE-C). We further include the recent TTS model E2-TTS (Eskimez et al., 2024) (both its original ZeroShot version and a version fine-tuned on the VisualTTS dataset, denoted E2-TTS-tuned) in our comparison. See Appendix F for details.

**Evaluation Results.** Table 1 presents sound generation results on VGGSound benchmark, where VSSFlow matches SOTA performance. Versus Frieren, VSSFlow excels in sound quality and sound-visual semantic alignment, though slightly weaker in temporal alignment. Against LoVA, VSSFlow — with fewer parameters — matches sound quality and outperforms in temporal and semantic alignment. Table 2 shows the VisualTTS results on Chem and GRID benchmarks. VSSFlow achieves strong results in Spk.Sim., MCD, and LSE, demonstrating its effective capture of speaker traits and generation of lip-synced, high-fidelity speech.

**Findings.** For V2S generation, the experiments highlight three key factors that affect the performance: video representations, training data quality, and model paradigm. First, CLIP features yield robust semantic representations for sound quality and semantic alignment but lack temporality, addressable via features like CAVP or Synchformer. Second, high-quality text-to-sound data or frozen text-to-sound modules further enhance results. Third, flow/diffusion and mask models outperform autoregressive ones. For VisualTTS, the metrics validate that a simple unified model suffices for VisualTTS generation without a complex data preprocessing pipeline. However, the use of VAE-vocoder incurs detail loss, reducing UTMOS versus ground truth (as seen in GT-vocoder baselines).

### 4.5 CASE STUDY: SOUND-SPEECH JOINT GENERATION CAPABILITY

VSSFlow shows excellent performance in V2S and VisualTTS tasks, but lacks the inherent ability of sound-speech joint generation. We stimulate this capability by fine-tuning pretrained VSSFlow model with 170k synthetic sound-speech mixtures, created via randomized concatenation or overlay of data from VGGSound and LRS2. Fine-tuning the base model for only 26k steps (15 epochs) yields a model capable of jointly generating sound and speech aligned with visual content. We evaluate its out-of-domain performance using videos from Veo3 (Google, 2025). As shown in Figure 1(c), the model accurately generates a car driving sound and a police officer's speech, along with the braking sound in the background. The generated speech is precisely aligned with the character's lip movements, timbre (adult male), and emotional tone (slightly excited). We politely invite you to visit our project page[2] for more playable demos. These results demonstrate VSSFlow's capability to process video signals and transcripts simultaneously, and the capability to generate the environmental sound alongside clear and accurate speech jointly.

## 5 DISCUSSION AND CONCLUSION

This work presents a unified flow model integrating video-to-sound (V2S) and visual text-to-speech (VisualTTS) tasks, establishing a new paradigm for video-conditioned sound and speech generation. Our framework demonstrates an effective condition aggregation mechanism for incorporating

---

[2]Project Page: `https://vasflow1.github.io/vasflow/`

speech and video conditions into the DiT architecture. Besides, we reveal a mutual boosting effect of sound-speech joint learning through analysis, highlighting the value of a unified generation model. For future research, there are several directions that merit further exploration. First, the scarcity of high-quality video-speech-sound data limits the development of unified generative models. Additionally, developing better representation methods for sound and speech, which can preserve speech details while maintaining compact dimensions, is a critical future challenge.

ETHICS STATEMENT

This work adheres to the ICLR Code of Ethics, with all authors having read and acknowledged it during submission. Our research on VSSFlow does not involve human subjects directly. It relies totally on publicly available datasets for model training. However, we recognize potential ethical concerns related to the model's applications, including the risk of generating misleading or harmful content, such as deepfake audio that could exacerbate misinformation or privacy violations if misused. Besides, there may be limitations for the groups that appear relatively infrequently in the training data. To mitigate discrimination, bias, and fairness issues, we evaluated the model on diverse datasets to ensure better robustness across different settings and demographics.

REPRODUCIBILITY STATEMENT

To facilitate reproducibility, this paper provides detailed descriptions of the VSSFlow architecture (Section 3.2). data preprocessing steps (Appendix B) and splitting method (Appendix D), evaluation metrics (Appendix E and F), training procedures with additional implementation details (Section 4.1, Appendix D), extensive ablation results (Appendix C), etc. In addition, to facilitate the replication and further research by the academic community, we promise to release the training code and checkpoint weights in our project page as soon as possible.

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

# A   DURATION PREDICTOR

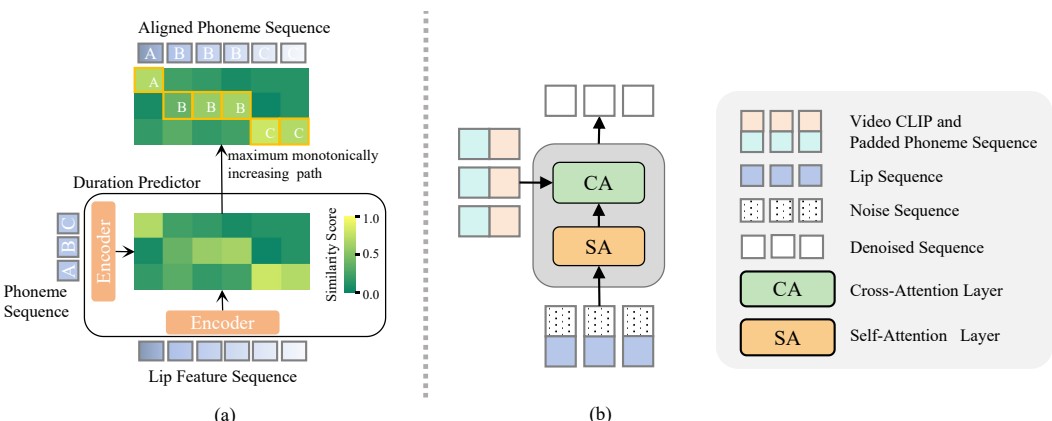

(a)                                      (b)

Figure 5:   (a) The architecture overview of our duration predictor. Specifically, the predictor employs a dual-encoder architecture to independently encode the phoneme and lip features. On top of these encoders, we compute the inter-sequence similarity between the encoded representations to generate a similarity matrix. (b) The architecture overview of the VSSFlow variant without explicit duration predictor. Lip features are concatenated as condition, phonemes are simply padded and introduced via cross-attention.

Inspired by prior works in text-to-speech (Chien et al., 2021; Cong et al., 2024a), we train a duration predictor to predict phoneme with lip movement. Our predictor takes phoneme and lip AV-HuBERT features (Shi et al., 2022) from video as inputs and outputs the similarity matrix between phoneme and lip feature. , as in Figure 5(a). During training, the model is constrained by an L1 loss between the predicted matrix and the ground-truth (GT) similarity matrix. During inference, based on the computed similarity matrix, we employ dynamic programming to compute the maximum monotonically increasing diagonal path in the matrix. We map each position on the optimal path to its corresponding phoneme. The predictor was trained for 300 epochs on the combined Chem, GRID, and LRS2 datasets using 4 GPUs, with a per-GPU batch size of 36 and a learning rate of 4e-7.

To better investigate this issue, we conduct additional experiments to verify whether the duration predictor is necessary for VisualTTS task. We train a model without any duration predictor. The model weight is initialized by the variant CrossV. Illustration about model architecture cen be found in Figure 5(b). Duration training, we adopt the same setting as before, which can be found in Appendix D. We compare the model (w/ duration predictor) and the variant (w/o duration predictor), the metrics are listed in table 3. For the model (w/ duration predictor), we use the groundtruth duration as guidance. However, VSSFlow without an explicit duration predictor even achieves significantly better synchronization metrics (LSE-C and LSE-D), with only minor sacrifices in WER and speaker similarity. This demonstrates that VisualTTS models, like modern TTS systems, can also benefit substantially from implicit duration modeling, thereby maintaining a simpler architecture while substantially improving overall performance.

Table 3:  VisualTTS evaluation results on Chem and GRID benchmark. For each metric, the highest score is in **bold** and the second-highest score is underlined.

| Bench | VSSFlow | WER ↓ | Spk. Sim. ↑ | UTMOS ↑ | MCD ↓ | MCD-D. ↓ | MCD-DS. ↓ | LSE-C ↑ | LSE-D ↓ |
|-------|---------|-------|-------------|---------|-------|----------|-----------|---------|---------|
| Chem | w/ DP | **13.1** | **80.7** | 3.2 | 10.04 | 5.48 | 5.49 | 6.08 | 8.44 |
|      | w/o DP | 13.3 | 74.6 | **3.32** | **8.23** | **4.94** | **4.94** | **7.73** | **6.85** |
| GRID | w/ DP | **16.1** | **55.9** | 3.24 | 8.02 | 5.8 | 5.81 | 6.53 | 8.44 |
|      | w/o DP | 16.8 | 49.6 | 3.24 | **8.93** | 6.01 | 6.01 | **6.76** | **8.22** |

## B CONDITION REPRESENTATIONS

Each video is truncated or padded to 10 seconds. For video representations, we employ CLIP (Radford et al., 2021) model to extract video features at 10 FPS, resulting in representations $c_v \in \mathbb{R}^{T_v \times D_v}$, where $T_v = 100$ is the total number of frames and $D_v = 768$ is the CLIP feature dimension. For speech representations, transcripts are first converted to phoneme sequences, which are then encoded by a custom-trained embedding network into representations $c_p \in \mathbb{R}^{T_p \times D_p}$. $T_p$ varies with sentence length and $D_p = 32$ denotes the phoneme embedding dimension.

Audio waveforms are resampled at 16kHz, truncated, or padded to 10 seconds. Waveforms are first converted into melspectrum and then encoded into a latent representation $x_1 \in \mathbb{R}^{T_a \times D_a}$ using the Variational Autoencoder (VAE) from AudioLDM 2 (Liu et al., 2024a), where $T_a = 250$ is the length of the latent sequence and $D_a = 64$ is the latent dimension. Timestep condition $t$ is encoded and padded before the latent representation sequence to form the final input latent $x_t \in \mathbb{R}^{(T_a+1) \times D_a}$ During inference, the predicted latent is converted into a mel-spectrogram through the VAE decoder and then reconstructed to a waveform by HiFiGAN vocoder (Kong et al., 2020a).

To ensure temporal alignment across modalities, video features $c_v$ are linearly interpolated to match the latent audio length $T_v = T_a = 250$, yielding the final $c_v \in \mathbb{R}^{250 \times 768}$. For the phoneme-based speech condition $c_p$, The predictor enables us to repeat and pad phoneme embeddings to obtain $T_p = T_a = 250$. This results in temporal-aligned phoneme features $c_p \in \mathbb{R}^{250 \times 32}$. To align with prior VisualTTS baselines, we extract speaker embeddings from reference speech using RawNet3 (Jung et al., 2022), optionally prefixing them to $c_v$ for speaker consistency.

## C EXPERIMENTS

### C.1 ABLATIONS ON CONDITION MECHANISM

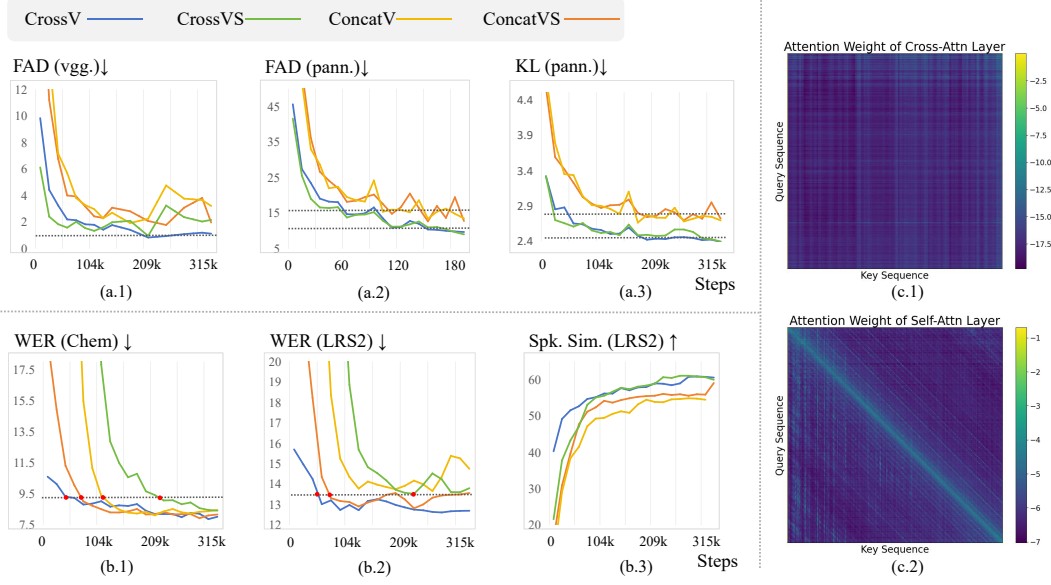

Figure 6: Performance comparison of models with different conditioning mechanisms over training steps. (a) Left-top three plots show metrics for the V2S task, while (b) Left-bottom three plots present metrics for the VisualTTS task. (c) Visualization of the attention weights in self- and cross-attention layers of DiT blocks.

## C.2 ABLATIONS ON JOINT TRAINING

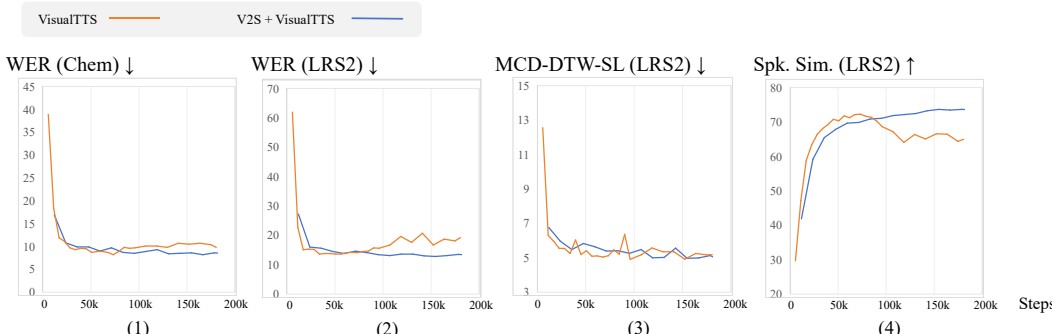

Figure 7: Impact of data configurations on VisualTTS generation performance. Two models are trained with VisualTTS data only and V2S + VisualTTS data. Speech generation metrics across training steps of the two models are plotted. Under the same training steps, both models achieve comparable performance, with only minor differences across evaluation metrics.

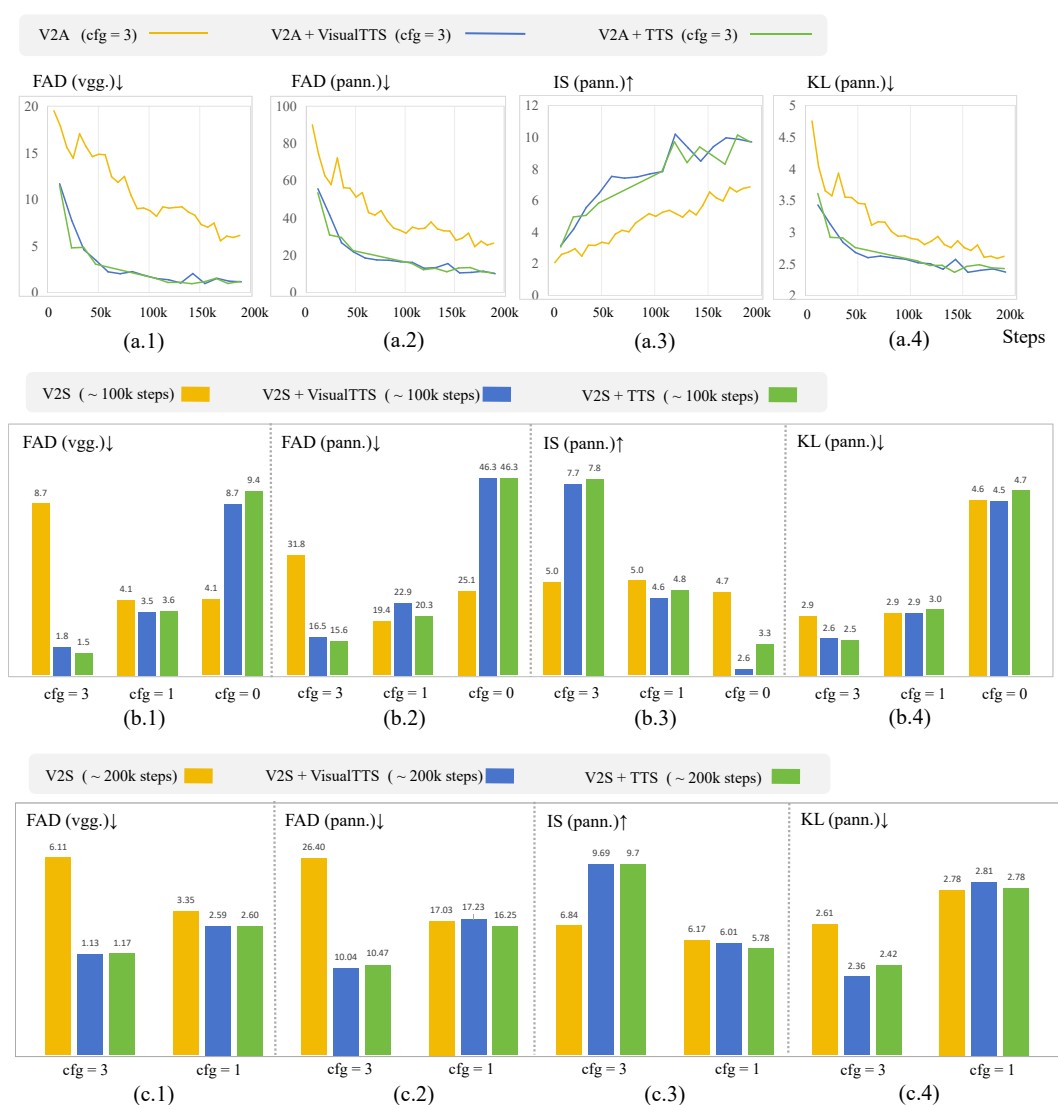

Figure 8: Impact of data configurations on V2S generation performance. Three models are trained with different data setups: V2S only, V2S + VisualTTS, and V2S + TTS. (a) The top four plots show sound generation metrics across training steps. (b) Bottom four plots compare the performance of models (trained on different data settings for ~100k steps) under varying classifier-free guidance (CFG) scales. While all models perform similarly under conditional generation (cfg = 1), those trained with additional speech data (+VisualTTS / +TTS) achieve significantly better performance when CFG is applied (cfg = 3).

## D DATASET AND EXPERIMENTS DETAILS

**V2S Dataset.** We use *VGGSound* (Chen et al., 2020), a widely adopted benchmark for the V2S task, which contains approximately **182k** training samples and 15k test samples. VGGSound provides a diverse collection of sound-visual pairs, making it well-suited for training models in video-conditioned sound generation.

**VisualTTS Datasets.** We employ three standard VisualTTS datasets: *Chem* (Prajwal et al., 2020), *GRID* (Cooke et al., 2006), and *LRS2* (Afouras et al., 2018), comprising a total of **162k** training samples. Chem is a single-speaker English dataset of chemistry lecture recordings. Following prior work (Cong et al., 2023), we use 6,240 samples for training and 200 for testing. GRID is a multi-speaker English dataset consisting of 33 speakers, each contributing 1,000 utterances. In line with (Cong et al., 2024a), we select 100 segments per speaker for testing, resulting in 29,600 training samples and 3,291 test samples. LRS2 is a diverse sentence-level dataset collected from BBC television programs, providing real-world variability. It contains 126k training samples and 700 test samples.

**TTS Datasets.** For the TTS task, we use *LJSpeech* (Ito & Johnson, 2017) and *LibriTTS* (Zen et al., 2019), totaling approximately **160k** training samples. LJSpeech is a single-speaker dataset with 13,100 narrated passages from nonfiction books. Following (Chien et al., 2021), we use 12,577 samples for training and 523 for testing. LibriTTS is a large-scale multi-speaker corpus of English read speech, comprising 147k training samples with diverse speaker identities.

During training, phoneme durations follow ground-truth alignments from speech data. We set the unconditional probability of video representation $c_v$ and speech representation $c_p$ to 0.1, use a batch size of 36 per GPU, a global learning rate of 4e-7, and apply 2,000 warm-up steps. The model is trained for 350k steps (200 epochs) on 4 H100 GPUs. At the inference stage, the Dopri5 is adopted to ensure high-quality sampling. We apply a classifier-free guidance scale of 3.0 for the V2S task and a scale of 1.5 for the VisualTTS task. For the joint generation case in Section 4.5, we use a scale of 3.0. Ablations of classifier-free guidance scale on V2S and VisualTTS tasks can be found in Table 4 and Table 5.

Table 4: Ablation on CFG scale for V2S task on VGGSound benchmark.

| CFG scale | FAD (vgg.) ↓ | FAD (pann.) ↓ | IS (pann.) ↑ | KL (pann.) ↓ |
|---|---|---|---|---|
| cfg = 1 | 3.55 | 18.88 | 4.99 | 2.95 |
| cfg = 3 | 1.34 | 11.10 | 9.48 | 2.44 |
| cfg = 5 | **1.24** | **10.88** | **11.09** | **2.43** |

Table 5: Ablation on CFG scale for VisualTTS task on Chem and GRID benchmark.

| Dataset | CFG scale | WER↓ | Spk Sim↑ | UTMOS↑ | MCD↓ | MCD-DTW↓ | LSE-C↑ | LSE-D↓ |
|---|---|---|---|---|---|---|---|---|
| GRID | cfg = 1 | 18.5 | 47.8 | 3.22 | **7.11** | **5.02** | **6.54** | **8.42** |
| | cfg = 3 | **16.1** | **55.9** | **3.24** | 8.02 | 5.80 | 6.53 | 8.44 |
| | cfg = 5 | 24.3 | 55.4 | 2.92 | 8.71 | 6.29 | 6.2 | 8.64 |
| Chem | cfg = 1 | 8.9 | 77.1 | 3.07 | 6.74 | **4.84** | 6.9 | 7.56 |
| | cfg = 3 | **8.1** | **81.8** | **3.39** | **6.73** | 4.97 | **7.13** | **7.45** |
| | cfg = 5 | 8.2 | 80.1 | 3.15 | 7.39 | 5.52 | 6.91 | 7.66 |

## E V2S BENCHMARK

**Baselines** We evaluate the video-to-sound performance on the standard VGGSound benchmark. We compare VSSFlow against baselines representing different paradigms, as shown in Table 1. Autoregressive baselines include SpecVQGan (Iashin & Rahtu, 2021), Im2Wav (Sheffer & Adi, 2022) and V-AURA (Viertola et al., 2025). Mask-based baselines include VAB (Pascual et al., 2024). Diffusion baselines include Difffoley (Luo et al., 2023), Seeing&Hearing (Xing et al., 2024), V2A-Mapper (Wang et al., 2024a), FoleyCrafter (Zhang et al., 2024), TiVA (Wang et al., 2024b)

and LoVA (Cheng et al., 2024). Flow-based approaches include Frieren (Wang et al., 2024c) and MMAudio (Cheng et al., 2025). The baseline results are obtained either through official code execution or from released generated sounds. All sound samples are padded to 10 seconds for consistency. Since V-AURA (Viertola et al., 2025) generates 2.56-second clips, we repeat each generated clip three times before padding it to 10 seconds.

Our primary comparisons focus on two baselines: **Frieren**(Wang et al., 2024c), which has a comparable parameter count, uses flow matching, and is trained on the same V2S dataset as VSSFlow. It introduces video conditions by concatenating video representations with latent. And **LoVA**(Cheng et al., 2024), which employs a 24-layer DiT backbone and adopts a diffusion paradigm. Compared to VSSFlow, LoVA has significantly more parameters and is fully initialized with Stable Audio Open weights.

**Metrics**   Consistent with prior work (Iashin & Rahtu, 2021; Sheffer & Adi, 2022; Luo et al., 2023; Liu et al., 2023), we adopt widely used metrics, including Fréchet Audio Distance (FAD)(Kilgour et al., 2018), Inception Score (IS)(Salimans et al., 2016), and Mean KL-Divergence (KL), to assess the quality of generated sound. To mitigate the impact of varying sampling rates, we first downsample the generated sound to 16 kHz and then resample it to the required rates for the respective audio classifiers (16 kHz for VGGish (Hershey et al., 2017), 32 kHz for PaSST (Koutini et al., 2022), and 32 kHz for PANN (Kong et al., 2020b)). For temporal alignment evaluation, we use Onset Accuracy (Onset Acc.) and Onset Average Precision (Onset AP), following (Zhang et al., 2024; Du et al., 2023). Additionally, inspired by (Cheng et al., 2025), we extract features using the SynchFormer model (Iashin et al., 2024) to compute offsets, producing the DeSync Score. To quantify soundvisual relevance, we calculate the cosine similarity between the embeddings of the input video and the generated sound using the ImageBind (Girdhar et al., 2023) model (VA-IB).

## F  VISUALTTS BENCHMARK

**Baselines**   We evaluate VSSFlow's performance against other VisualTTS baselines on the widely adopted Chem and GRID datasets. During generation, we randomly select a speech sample from the same speaker as the reference. We compare VSSFlow against four SOTA baselines: DSU (Lu et al., 2023), HPMDubbing (Cong et al., 2023), StyleDubber (Cong et al., 2024b) and EmoDubber (Cong et al., 2024a). We additionally include E2TTS and its fine-tuned version as a TTS baseline. It is fine-tuned on Chem + GRID dataset for 10k steps on 4 H800 GPUs with per-GPU batch size 32, learning rate 5e-5. Both the pretrained model weights and the training code are directly obtained from the github repository.[3]

The baseline data is either generated from the official implementation or from author-released results. We also obtain the results of the GT-vocoder by compressing the ground truth mel-spectrogram into the latent space via a VAE encoder, reconstructing it through a VAE decoder, and then using a vocoder to recover the waveform. Theoretically, the indicators in this row represent the upper limit for VSSFlow.

**Metrics**   We evaluate visual text-to-speech (VisualTTS) performance using several metrics. For overall speech quality, Word Error Rate (WER) measures intelligibility using the Whisper-V3 model (Radford et al., 2023), while the UTokyo-SaruLab Mean Opinion Score (UTMOS)(Saeki et al., 2022), computed via Versa(Shi et al., 2024), assesses clarity, naturalness, and fluency. Speaker similarity measures the similarity of two speech samples, also computed via Versa. For speech alignment between the ground truth speech and the predicted speech, Mel-Cepstral Distortion (MCD) quantifies the distance between two MFCC vectors. MCD-DTW applies Dynamic Time Warping (DTW) to compute the minimum MCD. MCD-DTW-SL, weighted by speech length, evaluates duration synchronization by incorporating a penalty term to capture local and global temporal similarities. For speech-visual alignment, Lip Sync Error Distance (LSE-D) and Lip Sync Error Confidence (LSE-C), computed using the pre-trained SyncNet (Chung & Zisserman, 2016) model, assess lip synchronization accuracy and confidence, respectively.

---

[3]`https://github.com/SWivid/F5-TTS`

## G    DISCUSSION

**Conclusion and future work.**    This work presents a unified flow model integrating video-to-sound (V2S) and visual text-to-speech (VisualTTS) tasks, establishing a new paradigm for video-conditioned sound and speech generation. Our framework demonstrates an effective condition aggregation mechanism for incorporating speech and video conditions into the DiT architecture. Besides, we reveal a mutual boosting effect of sound-speech joint learning through analysis, highlighting the value of a unified generation model. There are several directions that merit further exploration. First, the scarcity of high-quality video-speech-sound data limits the development of unified generative models. Additionally, developing better representation methods for sound and speech, which can preserve speech details while maintaining compact dimensions, is a critical future challenge.

**Limitation**    The use of synthesized sound-speech mixed data offers a viable approach to joint generation. However, its effectiveness is likely inferior to that of native sound-speech joint data. In future, when evaluation and data are sufficient, these synthesized data can be replaced to further enhance performance.

## H    THE USE OF LLMS

In this work, large language models (LLMs) were employed only as a writing assistant to refine drafts, improve clarity, and suggest structural improvements to the manuscript. However, all core research contributions—including model design, mathematical formulations, and experimental analyses—were developed independently by the authors. The authors take full responsibility for the final content, ensuring no plagiarism or fabrication occurred. This usage did not warrant authorship attribution to the LLM, as it remained auxiliary to human-led innovation.

