# OpenReview forum: "VSSFlow: Unifying Video-conditioned Sound and Speech Generation via Joint Learning"
_ICLR.cc/2026/Conference — Submitted to ICLR 2026_

### Official Review · Reviewer_o5uM · 2025-10-29

**Soundness:** 2
**Presentation:** 2
**Contribution:** 2
**Rating:** 6
**Confidence:** 4

**Summary:**

The paper introduces VSSFlow, a unified generative model that integrates the traditionally separate tasks of video-conditioned sound generation (V2S) and visual text-to-speech (VisualTTS) within an end-to-end FLOW-matching framework. It studies the condition aggregation mechanism within the DiT blocks and figures out how to effectively handle distinct input signals by leveraging cross-attention for ambiguous video conditions and self-attention for more deterministic speech transcripts. The authors also demonstrate that the joint learning of sound and speech generation helps faster convergence, better conditional generation, and allows VSSFlow to outperform state-of-the-art domain-specific baselines on both V2S and VisualTTS benchmarks.

**Strengths:**

- VSSFlow integrates V2S and VisualTTS tasks within a single end-to-end FLOW-matching framework
- VSSFlow outperforms existing domain-specific baselines on both V2S and VisualTTS benchmarks
- The paper does a comprehensive study of various condition aggregation mechanisms

**Weaknesses:**

- While the overall observation for the joint learning is interesting, all studies of conditioning design are healthy, and the generated samples sound great, the technical contribution is relatively incremental.
- It would be helpful to conduct a subjective evaluation to compare the generated samples against the baselines.
- The performance of V2S generation is affected by the choice of video representations. While the chosen CLIP features yield robust semantic representations for sound quality and semantic alignment, they lack temporality. VSSFlow is slightly weaker in temporal alignment compared to certain baselines like Frieren.

**Questions:**

- One simple question I have is, could the authors provide insights into how the flow-matching (as opposed to diffusion or autoregressive models) specifically facilitates the learned general audio prior that drives this mutual enhancement?
- Given that other time-aware features (CAVP or Synchformer) are suggested for future work, were any ablation studies performed using these temporal features during the development of the condition aggregation mechanism, or was the focus exclusively on optimizing the integration of CLIP features?
- The authors mentioned that the effectiveness of using synthesized sound-speech mixed data for joint generation is likely inferior to that of native sound-speech joint data. Could you elaborate a little bit more about this? I am curious if it is true that a limitation for future research and development of unified generative models is the scarcity of high-quality video-speech-sound data

---

> ### Author Response · Authors · 2025-11-24
> **Re1: Incremental Contributions (In response to Weaknesses.1)**
>
> Thanks for the feedback and for the very thoughtful questions throughout the review process! To our knowledge, **the mutual facilitation between sound and speech under fully end-to-end joint training has indeed been underexplored.** We hope our findings offer a fresh and valuable insight into the interplay between these two seemingly distinct audio modalities, and we are excited to see future work build on this direction.

---

> ### Author Response · Authors · 2025-11-24
> **Re2: Video Feature (In respond to Weakness.3)**
>
> Thank you for the insightful questions regarding our choice of video features. CLIP features possess strong semantic modeling capabilities and are therefore widely adopted in many Video-to-Speech (V2S) models. However, as pointed out by the reviewers, CLIP features provide less explicit temporal modeling than CAVP and Synchformer features. To directly address this concern, we have launched a new training run that replaces CLIP features with CAVP features. Training is expected to complete in ~2 days, and we will immediately update the evaluation result.

---

> ### Author Response · Authors · 2025-11-24
> **Re3: Flow, Diffusion and AR (In response to Questiones.1)**
>
> Thank you for the insightful question.
>
> ### 1. Flow vs. Diffusion
>
> Recent single-modality generation tasks consistently show that flow-matching converges faster and to a better optimum than diffusion under identical compute and architecture. Since jointly modeling speech + sound is significantly harder than single-modality generation, we deliberately choose the paradigm that already dominates in the easier single-task setting, expecting it to scale better to the harder joint task.
>
> ### 2. Flow vs. Autoregressive:
>
> Flow naturally models continuous signals and continuous latent spaces. Speech and environmental sound can be compressed into the same continuous latent space with a single VAE, enabling seamless interaction between the two modalities from the very first encoding stage.
>
> However, AR models rely on discrete audio codecs. For joint speech+sound, this forces two suboptimal choices: (i) two separate codecs (used in the concurrent DualDub), which increases complexity and prevents acoustic-level interaction, or (ii) one unified codec, which inevitably trades off reconstruction quality between speech and sound and remains immature. Even DualDub finally appends a flow-based refinement stage after its AR model to recover quality, indirectly acknowledging that AR alone is insufficient for high-fidelity mixed audio.
>
> In short, flow-matching offers the cleanest and most effective technical path: it simplifies the architecture, avoids tricky discrete codec trade-offs, and lets us focus purely on the core interaction between speech and sound. This is why we believe flow-matching is the key enabler for the observed mutual enhancement in our unified model.

---

> ### Author Response · Authors · 2025-11-24
> **Re4: The limitation of Synthesized Sound and Speech data (In response to Questions.3)**
>
> Thank you for this insightful comment.
>
> Recent impressive works such as ControlAudio have convincingly demonstrated that carefully synthesized sound-speech mixtures can already drive unified generation models to remarkable performance, which underscores the tremendous value of synthetic data for this task.
>
> In addition, synthetic data still struggles to capture the subtle that naturally arise between speech and environments sound. Classic examples include the Lombard effect (speakers unconsciously raising their voice in noisy settings such as bars), reverberation changes in open spaces (e.g., echo in valleys or canyons), and many other context-dependent prosodic and spectral adaptations. These delicate interplay phenomena are difficult to synthesized with current tools and represent an important and exciting direction for future data creation and modeling efforts.

---

### Official Review · Reviewer_Xsf8 · 2025-10-31

**Soundness:** 2
**Presentation:** 3
**Contribution:** 2
**Rating:** 4
**Confidence:** 4

**Summary:**

This paper proposes VSSFLOW, a unified model for video-to-sound (V2S) and visual text-to-speech (VisualTTS) generation.
The model introduces a conditional architecture: cross-attention over visual features, and concatenation with audio features followed by self-attention over transcripts.
This design enables efficient joint training on mixed V2S and visual TTS datasets without a multi-stage training process.
Experiments show that VSSFLOW outperforms existing domain-specific baselines on both V2S and VisualTTS benchmarks.

**Strengths:**

- The unification of V2S and VisualTTS is timely and relevant, addressing the need for a general sound generation across modalities.
- The model design is simple yet well-motivated, combining visual and textual conditioning in a natural and interpretable manner.
- The attention pattern analysis provides helpful insight into how multimodal dependencies should be captured within the network.
- The proposed method shows promising results across two distinct domains, demonstrating the potential of shared training.

**Weaknesses:**

1. **Benchmark inconsistencies**
- The reported FAD scores for Frieren differ substantially from both the original paper and prior works (e.g., MMAudio). It would strengthen the paper if the authors could clarify the cause of these discrepancies, as they directly affect the interpretation of the benchmark results.

2. **Choice of baselines**
 - The exclusion of V2A-M, FoleyC, and MMAudio from comparison is not sufficiently justified. For V2A-M and FoleyC, initializing from pre-trained text-to-sound modules is a fair and efficient approach. Also, for MMAudio, it offers smaller variants (e.g., 157M parameters) comparable to VSSFLOW. These baselines should be fairly included as baselines.
 - Although the paper mentions several unified models (Veo3, AudioGen-Omni, DeepAudio, DualDub), these are not directly compared. A brief quantitative or qualitative comparison would clarify the contribution.

3. **Unclear mutual benefits of joint training**
 - In Fig.7, the conditional and unconditional generation results (cfg=0 vs. cfg=1) for the V2S-only model are nearly identical, suggesting potential training instability. The reviewer encourages the authors to analyze whether the conditional learning mechanism failed in the V2S-only setting.
 - The reported CFG instability (cfg=3) seems inconsistent with prior V2S works (e.g., Frieren). Clarifying whether this is due to data composition or training dynamics would improve the analysis.
 - In Fig.4(b), the joint V2S + VisualTTS setup does not clearly outperform VisualTTS-only.

4. **Lack of subjective evaluation**
 - The paper reports only objective metrics. While these are standard, adding even a small-scale user study (e.g., a human preference or MOS study) would make the results more interpretable and provide a clearer understanding of the generation quality.

**Questions:**

- Why do some training curves in Figs. 4, 6, and 7 stop around 10k iterations? Did training collapse or terminate early?
- Have the authors compared VSSFLOW with an independent V2S and VisualTTS generation pipeline? This would help isolate the benefit of unification.
- Could the authors provide direct comparisons with prior unified models?
- Would it be feasible to include even a small-scale user evaluation to complement the objective metrics?

---

> ### Author Response · Authors · 2025-11-24
> **Re1: Clarify for Benchmark Inconsistencies and the Choice of Baselines (In response to Weaknesses.1 and Weaknesses.2)**
>
> We sincerely thank you for raising this important point about the evaluation setup.
> ### 1. Investigation into the causes of discrepancies in evaluation results
> Our evaluation code is implemented based on AudioLDM evaluation repository [https://github.com/haoheliu/audioldm_eval/tree/main/audioldm_eval], while MMAudio uses AV-Benchmark repository [https://github.com/hkchengrex/av-benchmark]. **After a thorough side-by-side audit of our code against the AVBenchmark repo, we identified the following key discrepancies:**
> - **Audio Normalization**: In our pipeline, we apply mean subtraction normalization to the input audio spectrograms, as implemented in the AudioLDM eval repo (see line 160 of [https://github.com/haoheliu/audioldm_eval/blob/main/audioldm_eval/datasets/load_mel.py]).
> - **Output Padding**: As noted in Appendix E (Baselines), to align with the standard 10-second test set clips in both repositories, we pad all generated audio from our models (and baselines) to exactly 10 seconds using zero-padding at the end.
> - **Resampling Method**: Different baselines output at varying rates (e.g., 44.1 kHz vs. 16 kHz). As noted in Appendix E (metrics), to minimize information loss from mismatched downsampling and upsampling during metric computation, we follow the protocol in LoVA[1]: all test set audio and baseline generations are first downsampled to 16 kHz, then upsampled to the specific rate required by each classifier (e.g., 32 kHz for PaSST).
>
> We re-evaluated both our VSSFlow model and Frieren using the AV-Benchmark pipeline (with its default preprocessing or with above data processing pipeline). The results are shown in the new Table below. AVB means the official evaluation code in AV-Benchmark, while AV-p stands for the AV-Benchmark evaluation code with the above pipeline.
>
> | Evaluation Method | Ours            |  Ours  |   Ours     | AVB-p | AVB-p  |  AVB-p  | AVB | AVB   |  AVB   |
> |----------|----------------|--------------|---------------|----------------|--------------|---------------|----------------|--------------|---------------|
> | Method   | FAD (vgg.) ↓ | FAD (pann.) ↓ | FAD (passt.) ↓ | FAD (vgg.) ↓ | FAD (pann.) ↓ | FAD (passt.) ↓ | FAD (vgg.) ↓ | FAD (pann.) ↓ | FAD (passt.) ↓ |
> | VASFlow  | 1.34           | 11.10        | 187.40        | 1.30           | 11.17        | 188.91        | 0.81       | 11.26    | 276.49    |
> | Fririen  | 2.09           | 29.13        | 214.59        | 2.08           | 29.08        | 214.58        | 1.44           | 12.56    | 220.99        |
>
> ### 2. Choice of baseliens
>
> Baselines that are trained with high-quality T2A data or leverages **frozen pretrained TTA** module are not directly comparable, as they tend to have better evaluation result. Compared to V2S data, Text-to-Audio data are significantly higher-quality, well-captioned and contains less noise. To avoid any confusion: models that start from a **TTA-initialized checkpoint but are fine-tuned on public V2S dataset (e.g. , TiVA) are fully comparable to ours**, and we do include them in all reported benchmarks (see Table 1).
> During training, we only used the widely adopted VGGSound dataset for the V2S  task. **Our main baseline, Fririen and LoVA , also uses V2S data only, ensuring the fairness of the comparison.**
>
> ### Citation
>
> [1] Cheng et al. LoVA: Long-Form Video-to-Audio Generation. ICASSP, 2025.

---

> ### Author Response · Authors · 2025-11-24
> **Re2. Refine the Experiments of Sound and Speech Joint Learning (In response to Weaknesses.3(3) and Question.1)**
>
> ### 1. Why do models with different data sizes have different numbers of training steps in Figure 4?
>
> We aligned only the number of training epochs (all models were trained for 160 epochs), not the total training steps. For sound generation task task, the V2S data (VGGSound) is updated exactly the same number of times (160 epochs) in all settings. Since different data settings have different data sizes (V2S, V2S + TTS and V2S + VisualTTS), training for the same number of epochs naturally results in different total steps. Models with additional speech data take more steps to complete 160 epochs. In the updated version we report results with fixed total steps to ensure fair comparison.
>
> ### 2. Aligning training steps of Joint Learning experiments in Figure 4
>
> During the rebuttal phase, we have **extended the yellow curve ( V2S dataset only) and orange curve (VisualTTS dataset only) to 200k steps (320 epochs) to ensure same training steps.** The mixed-data curves (blue and green) remain at 160 epochs as originally reported. The updated training curves are updated in the paper **Figure 4** and **Appendix C**. The new experiments confirm that our conclusions remain unchanged when aligning the training steps.
>
> ### 3. Analysis for the results in Figure 4
>
> From the updated curves, **the conclusions are even clearer and stronger: The joint training of sound and speech does not exhibit the previously observed mutual suppression; instead, it promoted each other.**
> For the sound-generation task, even though the V2S-only model (yellow) sees the VGGSound data far more times at any given step count, its performance and convergence speed remain markedly inferior to both the V2S + VisualTTS (blue) and V2S + TTS (green) variants. **This demonstrates that speech-related data is beneficial for the learning process of sound generation.**
> For the speech-generation task, adding a large amount of sound data (most of which contains no linguistic content) does not hurt, and in fact helps. The VisualTTS + V2S model (blue) exhibits significantly better scalability: it continues to improve with more training steps, whereas the VisualTTS-only model (orange) starts to overfit and degrade after roughly 100k steps. Thus, **the extra sound data effectively mitigates overfitting on the speech task,** even achieves higher Spk. Sim. and lower WER as shown in Fighre 4(b.2) and Appendix C.2.
>
> ### 4. Additional experiments investigating overfitting alleviation
> To further investigate whether this overfitting alleviation is caused by the additional sound data itself or by the model learning the conditional V2S task, **we have trained a new experiment that only leverages sound data only but not influenced by V2S task.  We train it on VisualTTS dataset plus unconditional V2S data (i.e., sound clips with their video conditioning deliberately blanked out, forcing the model to learn the unconditional distribution of sound).** This run is ongoing and expected to finish in approximately five days. We will immediately report the full training curves and final metrics, comparing the results from VisualTTS-only (orange curve in Figure 4)，VisualTTS + V2S data （the additional setting, with video feature blanked out），VisualTTS + V2S data + V2S Task (blue curve in Figure 4).

---

> ### Author Response · Authors · 2025-11-24
> **Re3: Explanation of the Observed CFG Phenomenon  (In response to Weaknesses.3(1) potential training instability and Weakness.3(2) CFG instability)**
>
> ### 1. Why FAD are nearly identical when CFG = 0 and CFG = 1?
>
> Thank you for this very sharp observation. We would clarify that, **metrics FAD and IS are dataset-distribution-level metrics: they compute statistical indicators over the entire test set**, rather than pairing each generated sample to its specific conditioning video. Therefore, when the models are trained on a certain dataset, it is possible that the conditional (CFG=1) and unconditional (CFG=0) FAD and IS are close.
>
> ### 2. Diagnosing of training instability
>
> To rigorously assess whether the model truly exploits the visual condition on a per-sample basis, the most informative metric is the paired **KL divergence reported in Figure 8(b.4).** Unlike FAD/IS, KL is computed pairwise: for every test video, we measure KL between the ground-truth audio and the generation audio. **The figure clearly shows that, conditional generation (CFG=1) yields substantially lower per-sample KL than unconditional generation (CFG=0), confirming that the model effectively uses the video condition at the individual-sample level.**
>
> ### 3. Compared to prior work, why CFG fail under our V2S-on setting?
>
> The primary factor governing CFG stability is the training steps and convergence situation of the V2S model. A very recent analysis [1] confirms that CFG performance degrades when the model is insufficiently trained, exactly the situation we observe in Figure.4(a.1) with the V2S-only variant after only **100k** steps. As can be seen from the graph, the model has not yet converged at this point. In contrast, current V2S systems such as Frieren has been trained for a much more steps **(1.3M steps in the first stage + 600k reflow + 500k distillation steps)**, ensuring full convergence and stable CFG behavior.
>
> ### 4. Why CFG works in V2S + TTS / VisualTTS setting?
>
> The conditional and unconditional modeling capabilities are both of crucial importance for the CFG. In the paper [1], "When the model is underfitted, a mismatch exists between the conditional and unconditional predictions during sampling, causing CFG to guide the sample in a direction that deviates significantly from the optimal trajectory."
>
> On the sound generation task, the V2S-only model (trained solely on VGGSound) performs noticeably worse than the mixed V2S + Speech models, even though it sees the sound data far more times within the same step budget. The core reason may lie in high heterogeneity of the sound data. When using V2S data only, learning a stable unconditional overall data distribution is slow. Speech data, by contrast, is far more structured and easier to model. **Adding speech therefore accelerates convergence to a high-quality unconditional overall audio data distribution after only ~100k steps, which enhanced the effectiveness of CFG.** This better unconditional model then enables effective CFG guidance that successfully exploits shared knowledge between sound and speech.
>
> ### 5. Further proof of the above finding
>
> Figure 4(a.2) illustrates this phenomenom clearly: At CFG = 1.0 (purely conditional generation), the three data settings (V2S, V2S + TTS, V2S + VisualTTS) perform similarly. This indicates that the conditional modeling capabilities of the three models are found to be quite similar. However at CFG = 3.0 (when the model’s unconditional modeling capability is also taken into account), the V2S-only model completely fails whereas V2S + TTS and V2S + VisualTTS improve substantially and remain stable.
>
> **Within our significantly lighter training regime, simply mixing speech data into training provides the model a strong, structured learning signal that accelerates convergence by orders of magnitude.** As a result, the V2S + Speech models achieve stable and effective CFG guidance after the same 100k steps at which the V2S-only model still collapses. This is also one of the core contributions of our work: an end-to-end flow-matching framework enables sound and speech data to mutually reinforce each other, eliminating the need for the extremely long  training steps.
>
> ### Citation
> [1] Fan et al. CFG-Zero*: Improved Classifier-Free Guidance for Flow Matching Models. arXiv:2503.18886, 2025.

---

> ### Author Response · Authors · 2025-11-24
> **Re4: Evaluation for Sound and Speech Joint Generation (In response to Questions.2 and Questions.3)**
>
> ### 1. Key challenges in Video-conditioned Sound and Speech joint generation
>
> We fully recognize that the community currently lacks publicly available, large-scale, real-world video datasets containing videos, rich environmental sounds and clear foreground speech, as well as easy-to-use benchmarks for the joint sound-speech generation task. The mentioned unified model, AudioGen-Omni and DeepAudio are unfortunately not open-sourced as well as do not support joint visual-to-sound-and-speech generation. Another model, DualDub is indeed an excellent concurrent work, but the authors have not released full checkpoints, making comparison currently infeasible. Because of this gap, it is unfortunately not yet possible to perform large-scale training or fully standardized evaluation on real data for the unified setting.
>
> ### 2. Our main foucs
>
> In this paper, our core contribution lies in (1) proposing a unified architecture that achieves performance comparable to dedicated specialist models on both V2S and VisualTTS benchmarks, (2) systematically studying architectural choices and task-fusion patterns that make such unification possible and effective, and (3) thoroughly investigating the mutual influence and interaction between sound and speech generation during joint training. In Section 4.5, we constructed synthetic sound and speech mixed data for the training of video-conditioned audio-speech joint generation. It is intended as a case study to demonstrate that VSSFlow can serve as a powerful unified base model capable of generating both naturalistic sound events and highly intelligible speech from corresponding visual inputs.
>
> ### 3. Our data synthesis pipeline
>
> **We randomly sample an environmental sound waveform and a speech utterance, then merge them in two designed ways**: 1. **Add mode (speech overlaid on sound background)**: We randomly pick a start time t in the sound clip and add the speech waveform into the interval [t, t + t_speech] with a random SNR. For efficiency, video conditions are merged in feature space: the same interval of sound-derived video features is mixed with the speech-derived video features using the same weighting. 2. **Insert mode (speech replacing a segment of sound)**: We directly overwrite the waveform interval [t, t + t_speech] with the speech utterance and correspondingly replace the video features in that interval with the speech video features. The code are provided in our anonymous repo https://github.com/vasflow1/vssflow/blob/main/dataset/video_mix.py.
>
> ### 4. Evaluation of the model’s capabilities on video-conditioned sound-speech joint generation
>
> We fully understand the reviewers’ desire for more rigorous and standardized comparisons. During the rebuttal period, we will further strengthen the on model's joint generation ability. **Currently, The readily available video-sound-speech data and easy-to-use benchmark is extremely scarce.** Therefore, we plan to use Veo-3 to synthesize a small evaluation set containing natural videos with both environmental sounds and clearly spoken foreground speech. This will yield a realistic mixed sound + speech test set of reasonable size (~20–30 clips). We will then generate audio for these videos using our VSSFlow (single unified model) and pipeline baselines. All generations will be evaluated by human. We are pushing this evaluation at the high priority and expect to complete the test set creation, generation, and full subjective results before the final rebuttal deadline.
>
> ### 5. Future Work
>
> In future work, we will focus on constructing and publicly releasing a large-scale, high-quality joint video-sound-speech dataset along with standardized benchmarks.

---

> ### Comment · Reviewer_Xsf8 · 2025-11-25
>
> Thank you for the additional experiments and explanations.
>
> Re1:
> Thank you for clarifying the differences in the evaluation pipelines.
> However, this clarification actually highlights the bias introduced by post-processing (audio normalization, output padding, or resampling).
> Comparing AVB and AVB-p, FAD (passt.) improves drastically for VSSFlow, whereas no such improvement is observed for Frieren. Similarly, FAD (pann.) degrades drastically for Frieren, whereas this does not occur for VSSFlow.
> According to the authors’ explanation, these discrepancies arise purely from post-processing and do not reflect the actual quality of the generated samples.
> Given this, can the authors justify why this evaluation protocol should still be considered fair?
>
> Re2:
> Thank you for clarifying the setting in Fig. 4. However, the revision does not adequately address W.3(3).
> Based on the updated figures and explanation, training solely on VisualTTS converges around 70–80k steps and then begins to overfit (WER and spk. sim. gradually worsen afterward).
> This suggests that training only on VisualTTS is more computationally efficient while achieving comparable performance, as early stopping can be applied before overfitting occurs.
> Therefore, the advantage of jointly using VisualTTS and V2S datasets for VisualTTS tasks remains unclear.
>
> Re3:
> Thank you for elaborating on the observed CFG phenomenon.
> However, the explanation contains a critical contradiction.
> The authors argue that CFG fails because the unconditional generator underfits the data.
> Nevertheless, in Fig. 8(b), the CFG = 0 condition (i.e., unconditional generation) yields better FAD scores for V2S than for V2S + VisualTTS or V2S + TTS. This indicates that the models trained on V2S + VisualTTS or V2S + TTS underfit or fail to converge, not the V2S-only model.
> As noted, FAD measures distribution-level distance, and higher scores indicate greater convergence with the data distribution.
> Given this, the current explanation does not clearly describe the mechanism by which joint training offers benefits.
>
> Re4:
> Thank you for clarifying the prior work on speech–sound joint generation and the overall motivation.
> I appreciate the author's effort to include Veo-3's results.

---

### Official Review · Reviewer_STq9 · 2025-10-31

**Soundness:** 3
**Presentation:** 3
**Contribution:** 2
**Rating:** 2
**Confidence:** 4

**Summary:**

This work presents the VSSFlow framework, which unifies three tasks: video-to-sound, text-to-speech, and visual TTS.
For me, the major contributions/highlights are:
(1) multimodal information aggregation strategy, as in sections 3.2 and 4.2;
(2) The synergy effect when training the three tasks together.
(3) the joint speech-sound generation capability with fine-tuning

**Strengths:**

(1) The presentation is clear; the literature coverage is sufficient.
(2) The reported quantitative results in Table 1 (V2S) and Table 2(VisualTTS) are competitive in most metrics.

**Weaknesses:**

For the condition aggregation mechanism (Sections 3.2 & 4.2)
(1) The proposed model is trained on limited data (<1M samples). The comparison of the structure designs is not convincing at this data scale. Empirically, I'm expecting the difference among these 4 designs would become marginal with more data available. I understand that more data may not be feasible during the review stage, so I would recommend the authors to use 1/2, 1/4, 1/8 of the original data volume to check if the performance gap among these structures would shrink with higher data volume.
(2) As in Fig 3.b, I wonder if the crossVS would give a similar WER provided sufficient computing, since the green line is still converging?
(3) Although the authors have conducted considerable experiments to compare these 4 structures, the underlying finding doesn't provide enough insight. Personally, I'm not convinced about the explanation simply with "inductive bias", "locality", etc, which doesn't establish a serious and strong logical connection to what is observed in experiments.

For the joint learning between sound and speech:
(1) For the experiments in Fig.4, I would recommend that the authors report the complete curves, rather than the current ones stopping in the middle, aka, finish the 20k step of training. e.g., for Fig.4.b.1, I don't know why the orange curve is terminated much earlier.
(2) Fig.4.b.2 indicates over-fitting. I would request the authors to justify: the alleviation of over-fitting in blue is because of the new task (visual TTS), rather than the additional data introduced by the new task.

For the joint speech-sound generation:
(1) Section 4.5 missed many details in its implementation. If the paper can provide a decent solution to this joint generation problem, it would be a great contribution. However, the details of how the fine-tuning data is simulated and how its quality is ensured are totally missing. Thus, I'm not fully convinced of this claim.
(2) Although the demo is good, I think the authors need a more rigorous justification of their method.

**Questions:**

(1) Your duration predictor accepts both text and video, and then frozen in training. How is that module pre-trained? I cannot find the details in the main content.
(2) When using concat, how do you sync/ensure the same length for phone and video? Can the poor performance of "concat" be attributed to the poor synchronization of these two information streams?
(3) Usually, TTS doesn't need a CFG coefficient, but sound generation does. You mention CFG in section 3.1, but you don't seriously mention this issue. Especially in the case of section 4.5, how do you use CFG in the joint scenario?
(4) Even the naive TTS with only text input would have WER generally < 10 in most cases, but yours is 18.2%; you also missed this naive baseline result. I suppose you need to show that the visual TTS is better than naive TTS, as more information is injected.

---

> ### Author Response · Authors · 2025-11-24
> **Re1: Architeature and Training Details of Duration Predictor. (In response to Questiones.1)**
>
> ### 1. The application of duration predictor in TTS and VisualTTS models.
>
> In conventional TTS system, duration is predicted from text only. With the advancement of technology, TTS systems have gradually phased out the explicit use of duration predictor. In VisualTTS, however, the core goal of duration predictor is to precisely align speech to lip. So it naturally takes both phoneme and lip feature sequences as input. In our work, the duration predictor is based on similarity matrix between lip features and phoneme features, aligning phoneme to match the movement of lip. We have update the details of our duration predictor and training process in the **Appendix A** and **Figure 5(a)**. Full code is also available in our anonymous repository https://github.com/vasflow1/vssflow/blob/main/worker/dp_tts.py.

---

> ### Author Response · Authors · 2025-11-24
> **Re2: The Impact of Scaling on Different Model Variants  (In respond to Weaknesses.1(1) and Weaknesses.1(2)**
>
> We carried out experiments to explore the impact of scaling on different variants of model: **as training steps, data scope, and model size increase, will the performance gap between different architectural variants (CrossV, CrossVS, ConcatV, ConcatVS) gradually diminish?** This is a classic and important concern nowadays. Compared to the design of model architectures, scaling is often a more straightforward and lower-risk approach to improving performance.
>
> ### 1. The Necessity of Research on Model Architecture
>
> We believe that studying the differences in conditioning mechanisms remains highly valuable for two key reasons: 1. Even with aggressive scaling, architectural advantages in both final performance and convergence speed often persist. A clear precedent is that DiT paper [1] showed that, at identical parameter counts, adaLN substantially outperforms cross-attention method. Therefore, adaLN are more widely adopted by numerous follow-up works (PixArt [2], MMDiT [3], MMAudio [4]) when injecting global conditions like timestep, etc. The research on different conditioning mechanism enables us to have more options when designing a better model. 2. Scaling is powerful but not applicable everywhere. In many practical scenarios — for example on-device deployment, low-resource languages, domain-specific adaptation, or real-time inference — both data scope and model size are severely constrained. In such regimes, architectures that deliver higher performance and faster training at modest scales become critically important. Thus, we believe architecture exploration is meaningful for both large-scale and low-resource contexts. **Therefore, we have included new experiments verifying the scaling laws of VSSFlow variants to ensure they adapt effectively to a wide range of resource constraints. We provide further scaling experiment on training steps, data scope, model size.**
>
> ### 2. Experiment of scaling on training steps
>
> For training steps, **we pushed all four variants to a much longer training schedule until 350k steps (200 epochs) in total, 2× the steps reported in the original submission.** The smoothed learning curves are shown in **Figure 3 and Appendix C.1** of the modified paper.
>
> The experiment in Figure 3 shows that, **as training steps increase, different model variants still exhibit performance differences.** The superiority of our proposed crossv structures is robust and does not vanish with significantly extended training. On the sound-generation track, CrossV and CrossVS consistently outperform ConcatV and ConcatVS throughout the entire 350k training steps. The gap does not shrink, confirming the sustained advantage of our proposed architecture. On the speech-generation track, convergence speed differs markedly: CrossV and ConcatVS already saturate around 80k steps, whereas CrossVS continues to improve and only reaches comparable performance of the others over 200k steps. Variant CrossVS shows a very slow convergence speed in speech generation task. The results in Appendix C.1 shows that, after 200 epochs of training, variant CrossVS still exhibits disadvantaged over the CrossV variant on both LRS2 and Chem dataset.
>
> ### 3. Experiment of scaling on data scope and model size (still ongoing)
>
> For data scope and model size, we compared the following two variants: CrossV and ConcatV. **They were trained using 1/2 of the original data volume and 1/2 of the original model size respectively.** These two ablation experiments are expected to be completed within 4 days, and we will immediately report the evaluation results.
>
> ### Citation
>
> [1] Peebles and Xie. Scalable Diffusion Models with Transformers. ICCV, 2023.
>
> [2]Chen et al. PixArt-α: Fast Training of Diffusion Transformer for Photorealistic Text-to-Image Synthesis. arXiv:2310.00426, 2023.
>
> [3] Esser et al. Scaling Rectified Flow Transformers for High-Resolution Image Synthesis. ICML, 2024.
>
> [4] Cheng et al. MMAudio: Taming Multimodal Joint Training for High-Quality Video-to-Audio Synthesis. CVPR, 2025.

---

> ### Author Response · Authors · 2025-11-24
> **Re3: Further Clarification on Differences Between Variants (In respond to Weaknesses.1(3))**
>
> To better explain the performance and convergence differences observed across our variants (CrossV, CrossVS, ConcatV, ConcatVS), we provide the following analysis. At its core, the performance gaps arise from the **fundamental mismatches in the alignment strength of different conditions (video features or phoneme sequences) and audio sequence**. Besides, the **inherent differences between sound and speech generation tasks also has an impact.**
>
> ### 1. Sound and speech generation exhibit significantly different learning difficulties.
>
> For the video-to-sound task, even under identical video frames, the space of plausible sound outputs is inherently broad and multimodal. For instance, footage of a car engine could map to vastly different roaring timbres, pitches, or intensities. Moreover, V2S datasets often include off-screen audio, ambient noise, or non-synchronized elements, leading to weak correspondence between a video frame feature (video) and the aligned waveform segment (waveform). **This results in high divergence in the conditional distribution P(waveform | video), making learning more challenging.**
> However in speech generation task,  the duration-aligned phoneme sequences provide a much stronger supervisory signal at each position. **Given a phoneme sequence, the resulting waveform primarily varies only in subtle details like timbre or prosody, yielding a far sharper conditional distribution P(waveform | phoneme).** This explains the markedly faster convergence of different model variants (e.g., in Figure 3(b), CrossV and ConcatVS converge at ~40k steps, v.s. CrossVS at ~210k steps).
>
> ### 2. Different Conditioning mechanisms exhibit clear differences when incorporating information
>
> **Self-attention excels at interactions within homogeneous data sequences, fostering local and spatial coherence, while cross-attention is designed for injecting heterogeneous signals with more flexible alignments.** Extensive prior work has illuminated these distinctions. A 2023 TVCG study [1] on Vision Transformers revealed that self-attention layers exhibit strong locality and spatial bias, concentrating on neighboring patches to capture structural patterns. Similarly, Liu et al. at CVPR2024 [2] analyses the text-to-image (T2I) models, and found that self-attention and cross-attention shows different spatial perception ability.
>
>
> ### 3. Therefore, different tasks are better suited to different conditioning mechanisms
> Hard-concatenating video features with audio sequences at corresponding positions (as in Variant ConcatV and ConcatVS) and processing via self-attention thus injects a global, fuzzy, and poorly aligned signal into every audio position. This create conflicts that hinder learning process and generation performance for  video-to-sound task. **Therefore, the video sequence is more suitable for cross-attention mechanisim to achieve more free interaction. For the phoneme sequence, concatenating it with the audio latent sequence at corresponding positions can provide more accurate guidance signals, thereby enabling faster convergence in the joint training process.**
>
>
> ### Citation
> [1] Li et al. How Does Attention Work in Vision Transformers? A Visual Analytics Attempt. IEEE TVCG, 2023.
>
> [2] Liu et al. Towards Understanding Cross and Self-Attention in Stable Diffusion for Text-Guided Image Editing. CVPR, 2024.

---

> ### Author Response · Authors · 2025-11-24
> **Re4. Refine the Experiments of Sound and Speech Joint Learning (In response to Weaknesses.2(1) and Weaknesses.2(2))**
>
> ### 1. Why do models with different data sizes have different numbers of training steps in Figure 4?
>
> We aligned only the number of training epochs (all models were trained for 160 epochs), not the total training steps. For sound generation task task, the V2S data (VGGSound) is updated exactly the same number of times (160 epochs) in all settings. Since different data settings have different data sizes (V2S, V2S + TTS and V2S + VisualTTS), training for the same number of epochs naturally results in different total steps. Models with additional speech data take more steps to complete 160 epochs. In the updated version we report results with fixed total steps to ensure fair comparison.
>
> ### 2. Aligning training steps of Joint Learning experiments in Figure 4
>
> During the rebuttal phase, we have **extended the yellow curve ( V2S dataset only) and orange curve (VisualTTS dataset only) to 200k steps (320 epochs) to ensure same training steps.** The mixed-data curves (blue and green) remain at 160 epochs as originally reported. The updated training curves are updated in the paper **Figure 4** and **Appendix C**. The new experiments confirm that our conclusions remain unchanged when aligning the training steps.
>
> ### 3. Analysis for the results in Figure 4
>
> From the updated curves, **the conclusions are even clearer and stronger: The joint training of sound and speech does not exhibit the previously observed mutual suppression; instead, it promoted each other.**
> For the sound-generation task, even though the V2S-only model (yellow) sees the VGGSound data far more times at any given step count, its performance and convergence speed remain markedly inferior to both the V2S + VisualTTS (blue) and V2S + TTS (green) variants. **This demonstrates that speech-related data is beneficial for the learning process of sound generation.**
> For the speech-generation task, adding a large amount of sound data (most of which contains no linguistic content) does not hurt, and in fact helps. The VisualTTS + V2S model (blue) exhibits significantly better scalability: it continues to improve with more training steps, whereas the VisualTTS-only model (orange) starts to overfit and degrade after roughly 100k steps. Thus, **the extra sound data effectively mitigates overfitting on the speech task,** even achieves higher Spk. Sim. and lower WER as shown in Fighre 4(b.2) and Appendix C.2.
>
> ### 4. Additional experiments investigating overfitting alleviation
> To further investigate whether this overfitting alleviation is caused by the additional sound data itself or by the model learning the conditional V2S task, **we have trained a new experiment that only leverages sound data only but not influenced by V2S task.  We train it on VisualTTS dataset plus unconditional V2S data (i.e., sound clips with their video conditioning deliberately blanked out, forcing the model to learn the unconditional distribution of sound).** This run is ongoing and expected to finish in approximately five days. We will immediately report the full training curves and final metrics, comparing the results from VisualTTS-only (orange curve in Figure 4)，VisualTTS + V2S data （the additional setting, with video feature blanked out），VisualTTS + V2S data + V2S Task (blue curve in Figure 4).

---

> ### Author Response · Authors · 2025-11-24
> **Re5: Evaluation for Sound and Speech Joint Generation (In response to Weaknesses.3(1) and Weakness.3(2))**
>
> ### 1. Key challenges in Video-conditioned Sound and Speech joint generation
>
> We fully recognize that the community currently lacks publicly available, large-scale, real-world video datasets containing videos, rich environmental sounds and clear foreground speech, as well as easy-to-use benchmarks for the joint sound-speech generation task. The mentioned unified model, AudioGen-Omni and DeepAudio are unfortunately not open-sourced as well as do not support joint visual-to-sound-and-speech generation. Another model, DualDub is indeed an excellent concurrent work, but the authors have not released full checkpoints, making comparison currently infeasible. Because of this gap, it is unfortunately not yet possible to perform large-scale training or fully standardized evaluation on real data for the unified setting.
>
> ### 2. Our main foucs
>
> In this paper, our core contribution lies in (1) proposing a unified architecture that achieves performance comparable to dedicated specialist models on both V2S and VisualTTS benchmarks, (2) systematically studying architectural choices and task-fusion patterns that make such unification possible and effective, and (3) thoroughly investigating the mutual influence and interaction between sound and speech generation during joint training. In Section 4.5, we constructed synthetic sound and speech mixed data for the training of video-conditioned audio-speech joint generation. It is intended as a case study to demonstrate that VSSFlow can serve as a powerful unified base model capable of generating both naturalistic sound events and highly intelligible speech from corresponding visual inputs.
>
> ### 3. Our data synthesis pipeline
>
> **We randomly sample an environmental sound waveform and a speech utterance, then merge them in two designed ways**: 1. **Add mode (speech overlaid on sound background)**: We randomly pick a start time t in the sound clip and add the speech waveform into the interval [t, t + t_speech] with a random SNR. For efficiency, video conditions are merged in feature space: the same interval of sound-derived video features is mixed with the speech-derived video features using the same weighting. 2. **Insert mode (speech replacing a segment of sound)**: We directly overwrite the waveform interval [t, t + t_speech] with the speech utterance and correspondingly replace the video features in that interval with the speech video features. The code are provided in our anonymous repo https://github.com/vasflow1/vssflow/blob/main/dataset/video_mix.py.
>
> ### 4. Evaluation of the model’s capabilities on video-conditioned sound-speech joint generation
>
> We fully understand the reviewers’ desire for more rigorous and standardized comparisons. During the rebuttal period, we will further strengthen the on model's joint generation ability. **Currently, The readily available video-sound-speech data and easy-to-use benchmark is extremely scarce.** Therefore, we plan to use Veo-3 to synthesize a small evaluation set containing natural videos with both environmental sounds and clearly spoken foreground speech. This will yield a realistic mixed sound + speech test set of reasonable size (~20–30 clips). We will then generate audio for these videos using our VSSFlow (single unified model) and pipeline baselines. All generations will be evaluated by human. We are pushing this evaluation at the high priority and expect to complete the test set creation, generation, and full subjective results before the final rebuttal deadline.
>
> ### 5. Future Work
>
> In future work, we will focus on constructing and publicly releasing a large-scale, high-quality joint video-sound-speech dataset along with standardized benchmarks.

---

> ### Author Response · Authors · 2025-11-24
> **Re6: Further Clarification on the Concatnation Method (In response to Question.2)**
>
> Our duration predictor ensures that the phoneme features and video features are aligned in time sequence. Please refer to more implementation details in **Appendix A** in our revised paper. Our duration predictor extends the phoneme features to the same length as the lip features, and the lip features have the same length as the video frames, thus ensuring that the lengths of phonemes and video frames are consistent. Furthermore, you mentioned “poor performance of concatenation”. Could you please specify which task you refers to?  We would be happy to provide more detailed analysis or additional experiments once we understand the exact point of concern. Thank you!

---

> ### Author Response · Authors · 2025-11-24
> **Re7: Experiments on CFG (In response to Question.3)**
>
> Thank you for the excellent question about CFG behavior. First of all,we would like to clarify that we mention the issue of CFG in Section 3.1. And we conducted a more detailed discussion on the effect of CFG in the joint training process **in Sections 4.3 and Figure 4(a.2)**.
>
> ### 1. Why does the effect of CFG become more obvious after adding speech data in sound generation task?
>
> CFG has become the default sampling enhancement method for both diffusion and flow-matching models, which leverages model’s ability capability in on conditional and unconditional generation. Via the experiment we conduct In in Figure 4(a.2) and Section 4.3 (Findings), we draw a key observation:  On the sound generation task, the V2S-only model (trained solely on VGGSound) performs noticeably worse than the mixed V2S + Speech models, even though it sees the sound data far more times within the same step budget. The core reason may lie in high heterogeneity of the sound data. When using V2S data only, learning a stable unconditional overall data distribution is slow. Speech data, by contrast, is far more structured and easier to model. **Adding speech therefore accelerates convergence to a high-quality unconditional overall audio data distribution after only ~100k steps. This better unconditional model then enables effective CFG guidance that successfully exploits shared knowledge between sound and speech.**
>
> Figure 4(a.2) illustrates this clearly: At CFG = 1.0 (purely conditional generation), the three data settings (V2S, V2S + TTS, V2S + VisualTTS) perform similarly. This indicates that the conditional modeling capabilities of the three models are found to be quite similar. However at CFG = 3.0 (when the model’s unconditional modeling capability is also taken into account), the V2S-only model completely fails whereas V2S + TTS and V2S + VisualTTS improve substantially and remain stable.
>
> ### 2. Ablation experiment on CFG scales for sound and speech generation task
> We additionally **ablated the CFG scale in sound and speech generation tasks.** We have updated them in the **Appendix D** and **Table 4-5.** For sound generation, a higher CFG within certain limits helps the generatuon. On speech generation, CFG = 3 gives clear gains on most metrics compared with CFG = 1, but performance drops beyond that point. Consequently, we use the same CFG = 3 in our joint sound + speech generation case study.
>
>
> | Dataset | CFG scale | FAD (vgg.) ↓ | FAD (pann.) ↓ | IS (pann.) ↑ | KL (pann.) ↓ |
> |-----------|-----------|--------------|---------------|--------------|--------------|
> | **VGGSound** | cfg = 1   | 3.55         | 18.88         | 4.99         | 2.95         |
> |                    | cfg = 3   | 1.34         | 11.10         | 9.48         | 2.44         |
> |                    | cfg = 5   | **1.24**     | **10.88**     | **11.09**    | **2.43**     |
>
>
>
>
> | Dataset | CFG scale | WER ↓ | Spk Sim ↑ | UTMOS ↑ | MCD ↓ | MCD-DTW ↓ | LSE-C ↑ | LSE-D ↓ |
> |---------|-----------|-------|-----------|---------|-------|-----------|---------|---------|
> | **GRID** | cfg = 1   | 18.5  | 47.8      | 3.22    | **7.11** | **5.02** | **6.54** | **8.42** |
> |         | cfg = 3   | **16.1** | **55.9** | **3.24** | 8.02  | 5.80   | 6.53  | 8.44  |
> |         | cfg = 5   | 24.3  | 55.4      | 2.92    | 8.71  | 6.29   | 6.20  | 8.64  |
> | **Chem** | cfg = 1   | 8.9   | 77.1      | 3.07    | 6.74  | **4.84** | 6.90  | 7.56  |
> |         | cfg = 3   | **8.1** | **81.8** | **3.39** | **6.73** | 4.97 | **7.13** | **7.45** |
> |         | cfg = 5   | 8.2   | 80.1      | 3.15    | 7.39  | 5.52   | 6.91  | 7.66  |

---

> ### Author Response · Authors · 2025-11-24
> **Re8: Add TTS baselines on VisualTTS benchmark (In response to Question.4: Missing TTS baselines in VisualTTS benchmark)**
>
> ### 1. The reason for high WER on GRID benchmark
>
> The GRID corpus contains a large number of highly unnatural, chess-related commands such as “BIN BLUE AT F TWO NOW”. Modern ASR systems may not perform well on such data, so they struggle on GRID, which explains the relatively high WER values. For reference, recent visualtts models that report on GRID all obtain WER > 15 (Emodubber [1]: 18.5, StyleDubber [2]: 18.8). Moreover, as shown in Table 2, **even the ground-truth human speech in GRID yields a WER of 12.9 with the same ASR model, confirming the intrinsic difficulty of the corpus.**
>
> ### 2. TTS baseline on VisualTTS task
>
> To provide a fairer reference, we have added E2TTS as an additional text-driven baseline.
> -  E2 TTS-zeroshot: direct inference with the official pretrained checkpoint
> -  E2 TTS-tuned: further finetuned on Chem + GRID for 10k steps (4× H800, per-GPU bs=32, lr=5e-5)
>
> Training details have been updated at Appendix F (Baselines). And TTS baselines results are updated in the VisualTTS benchmark results in the revised paper.
>
> | Dataset | Model                | WER ↓   | Spk Sim ↑ | UTMOS ↑ | MCD ↓   | MCD-DTW ↓ | MCD-DTW-SL ↓ | LSE-C ↑ | LSE-D ↓  |
> |---------|----------------------|---------|-----------|---------|---------|-----------|--------------|---------|----------|
> | Chem | E2TTS_ZeroShot  | 8.6 | 67.4   | 3.43| 14.73   | 5.89      | 7.21         | 1.50    | 13.17    |
> |         | E2TTS_Tuned          | 8.7 | 70.2  | 3.51| 14.46   | 5.07  | 6.21    | 1.62| 13.10|
> | GRID | E2TTS_ZeroShot  | 21.4    | 43.2      | 3.57    | 15.71   | 5.62      | 7.05         | 1.93    | 12.29    |
> |         | E2TTS_Tuned         | 20.2| 44.2  | 3.62| 15.64| 5.59  | 6.98    | 2.36| 12.08|
>
> ### 3. The performance difference of TTS and VisualTTS baselines
> As expected, the **TTS baselines perform well on WER and UTMOS** metrics because they are not constrained by lip synchronization. However, **on visually related metrics (LSE-D, LSE-C, and Spk-Sim), VisualTTS models significantly outperforms the TTS baselines, which have no access to visual input at all.**
>
> ### Citation
> [1] Cong et al. Emodubber: Towards High Quality and Emotion Controllable Movie Dubbing. CVPR, 2025.
> [2] Cong et al. StyleDubber: Towards Multi-Scale Style Learning for Movie Dubbing. ACL Findings, 2024.

---

### Official Review · Reviewer_Ssc7 · 2025-11-02

**Soundness:** 2
**Presentation:** 3
**Contribution:** 2
**Rating:** 4
**Confidence:** 3

**Summary:**

The paper introduces VSSFlow, a unified flow-matching model designed to handle both video-to-sound (V2S) and visual text-to-speech (VisualTTS) tasks within a single end-to-end framework. VSSFlow proposes a condition aggregation mechanism that leverages cross-attention for ambiguous video inputs and self-attention (concatenation) for deterministic text (transcript) inputs. The model demonstrates that joint training of sound and speech not only avoids interference but also yields mutual performance gains.

**Strengths:**

1. VSSFlow elegantly unifies V2S and VisualTTS tasks in a single flow-matching architecture, an open challenge that prior works (AudioGen-Omni, DualDub) only partially address.
2. The study provides clear empirical evidence that joint training across sound and speech modalities yields synergistic improvements—a finding that contrasts with previous assumptions of mutual degradation.

**Weaknesses:**

1. While the paper motivates Visual TTS as part of a unified audio-visual generation framework, the practical utility of video conditioning in TTS remains unclear. In realistic applications such as dubbing for AI-generated videos, lip features may not correspond accurately to the transcript, potentially introducing misalignment biases. The authors should clarify whether the learned representations rely on accurate lip motion, and how the model behaves when video features are noisy or inconsistent with speech content.
2. The paper provides an interesting analysis of different condition aggregation mechanisms (e.g., four variants), revealing inductive biases between ambiguous (visual) and deterministic (text) inputs. However, it remains unclear whether these observations scale with model capacity and data size. At larger scales, where multimodal representations are more expressive and robust, such design choices may have diminishing impact. It would strengthen the paper to discuss this potential limitation or provide scaling experiments.
3. The use of CLIP features at 10 FPS may be a limiting factor for temporally precise synchronization. The model’s slight underperformance in onset alignment (vs. Frieren) suggests potential weaknesses in temporal modeling.

**Questions:**

Recent TTS models (e.g., MaskGCT, E2-TTS, F5-TTS) have shown that explicit duration predictors can be removed without degrading alignment quality, leveraging implicit temporal modeling within the generative framework. In VSSFlow, the authors retain a traditional duration predictor for VisualTTS. Could the authors clarify why this design choice remains necessary, and whether their model could benefit from implicit duration modeling consistent with modern end-to-end TTS architectures?

---

> ### Author Response · Authors · 2025-11-24
> **Re1: Necessity of Duration Predictor and Robustness of Lip feature. ( In respond to Weaknesses.1 and Questions )**
>
> ### 1. The application of duration predictor in TTS and VisualTTS models.
>
> In conventional TTS system, duration is predicted from text only. With the advancement of technology, TTS systems have gradually phased out the explicit use of duration predictor. In VisualTTS, however, the core goal of duration predictor is to precisely align speech to lip. So it naturally takes both phoneme and lip feature sequences as input. In our work, the duration predictor is based on similarity matrix between lip features and phoneme features, aligning phoneme to match the movement of lip. We have update the details of our duration predictor and training process in the Appendix A. Full code is also available in our anonymous repository https://github.com/vasflow1/vssflow/blob/main/worker/dp_tts.py.
>
>
> ### 2. The robustness of lip features.
>
> To access the robustnes to AV-Hubert lip features, we randomly masked 5%, 10%, and 20% of the lip features at inference time. The evaluation results are listed in the following table.  As shown, **even though no specific training for robustness was applied to the duration predictor, it still exhibits robustness when the masking ratio is ≤10%.** Nearly all metrics remain essentially on par with published baselines, with no drastic degradation. Unsurprisingly, the lip-sync metrics (LSE-C and LSE-D) are the most sensitive and decline more visibly, which matches our intuition. However, when the masking ratio reaches 20%, we observe a clear drop across the board. This confirms that the accuracy of lip features directly affect the duration prediction quality and, which in turn affect the final speech intelligibility and lip-speech synchronization.
> | Mask Ratio | WER (↓) | Spk Sim (↑) | UTMOS (↑) | MCD (↓) | MCD-DTW (↓) | MCD-DTW-SL (↓) | LSE-C (↑) | LSE-D (↓) |
> |-|-|-|-|-|-|-|-|-|
> | 0%         | 15.1    | 79.7        | 3.17      | 9.55    | 5.18        | 5.19           | 6.10      | 8.37      |
> | 5%         | 17.1    | 78.6        | 3.09      | 10.23   | 5.32        | 5.32           | 5.65      | 8.53      |
> | 10%        | 19.5    | 79.5        | 3.05      | 10.47   | 5.40        | 5.40           | 5.47      | 8.99      |
> | 20%        | 27.7    | 78.4        | 2.91      | 12.31   | 6.18        | 6.18           | 4.36      | 10.14     |
>
> ### 3. Can we remove duration predictor in VisualTTS task?
>
> Nearly all VisualTTS baselines  (like Emodubber [1], StyleDubber [2])  rely on explicit duration modeling. While some very recent impressive works (e.g., AudioGen-Omni) remove the explicit duration predictor to simplify the architecture.  However, their reported metrics unfortunately do not include lip-sync scores such like LSE-C or LSE-D. Therefore, based on the current publicly available data, it is still difficult to determine whether the removal of dp will affect performance.  We fully appreciate the timely question whether explicit duration modeling can eventually be removed, as done in recent strong TTS systems.To better investigate this issue, we conduct additional experiments to verify it.
>
> **We have trained a model without any duration predictor: lip features are concatenated as conditioning, phonemes are simply padded and introduced via cross-attention.** More details can be found in Appendix A and Figure 5(b). Evaluations metrics are listed below. From the table, **VSSFlow without an explicit duration predictor even achieves significantly better synchronization metrics (LSE-C and LSE-D), with only minor sacrifices in WER and speaker similarity. This demonstrates that VisualTTS, like modern TTS systems, can also benefit substantially from implicit duration modeling, thereby maintaining a simpler architecture while substantially improving overall performance.** We sincerely thank the reviewer for this insightful comment — the additional ablation study greatly strengthens the paper.
>
> | Dataset | Method   | WER (↓) | Spk Sim (↑) | UTMOS (↑) | MCD (↓) | MCD-DTW (↓) | MCD-DTW-SL (↓) | LSE-C (↑) | LSE-D (↓) |
> |-|-|-|-|-|-|-|-|-|-|
> | **Chem**| w/ D.P.    |**13.1** |**80.7**| 3.20| 10.04| 5.48| 5.49| 6.08| 8.44|
> |           | w/o D.P.  | 13.3| 74.6 | **3.32**  | **8.23**| **4.94**| **4.94** | **7.73**| **6.85**|
>
> | Dataset | Method   | WER (↓) | Spk Sim (↑) | UTMOS (↑) | MCD (↓) | MCD-DTW (↓) | MCD-DTW-SL (↓) | LSE-C (↑) | LSE-D (↓) |
> |---------|----------|---------|-------------|-----------|---------|-------------|----------------|-----------|-----------|
> | **GRID**    | w/ D.P.    | **16.1**    | **55.9**       | **3.24**      | **8.02**    |**5.80**       | **5.81**           | 6.53          | 8.44      |
> |              | w/o D.P.   | 16.8        | 49.6              | **3.24**      | 8.93         | 6.01             | 6.01                 | **6.76**      | **8.22**     |
>
> ### Citation
>
> [1] Cong et al. Emodubber: Towards High Quality and Emotion Controllable Movie Dubbing. CVPR, 2025.
>
> [2] Cong et al. StyleDubber: Towards Multi-Scale Style Learning for Movie Dubbing. ACL Findings, 2024.

---

> ### Author Response · Authors · 2025-11-24
> **Re2: The Impact of Scaling on Different Model Variants (In respond to Weakness.2)**
>
> We carried out experiments to explore the impact of scaling on different variants of model: **as training steps, data scope, and model size increase, will the performance gap between different architectural variants (CrossV, CrossVS, ConcatV, ConcatVS) gradually diminish?** This is a classic and important concern nowadays. Compared to the design of model architectures, scaling is often a more straightforward and lower-risk approach to improving performance.
>
> ### 1. The Necessity of Research on Model Architecture
>
> We believe that studying the differences in conditioning mechanisms remains highly valuable for two key reasons: 1. Even with aggressive scaling, architectural advantages in both final performance and convergence speed often persist. A clear precedent is that DiT paper [1] showed that, at identical parameter counts, adaLN substantially outperforms cross-attention method. Therefore, adaLN are more widely adopted by numerous follow-up works (PixArt [2], MMDiT [3], MMAudio [4]) when injecting global conditions like timestep, etc. The research on different conditioning mechanism enables us to have more options when designing a better model. 2. Scaling is powerful but not applicable everywhere. In many practical scenarios — for example on-device deployment, low-resource languages, domain-specific adaptation, or real-time inference — both data scope and model size are severely constrained. In such regimes, architectures that deliver higher performance and faster training at modest scales become critically important. Thus, we believe architecture exploration is meaningful for both large-scale and low-resource contexts. **Therefore, we have included new experiments verifying the scaling laws of VSSFlow variants to ensure they adapt effectively to a wide range of resource constraints. We provide further scaling experiment on training steps, data scope, model size.**
>
> ### 2. Experiment of scaling on training steps
>
> For training steps, **we pushed all four variants to a much longer training schedule until 350k steps (200 epochs) in total, 2× the steps reported in the original submission.** The smoothed learning curves are shown in **Figure 3 and Appendix C.1** of the modified paper.
>
> The experiment in Figure 3 shows that, **as training steps increase, different model variants still exhibit performance differences.** The superiority of our proposed crossv structures is robust and does not vanish with significantly extended training. On the sound-generation track, CrossV and CrossVS consistently outperform ConcatV and ConcatVS throughout the entire 350k training steps. The gap does not shrink, confirming the sustained advantage of our proposed architecture. On the speech-generation track, convergence speed differs markedly: CrossV and ConcatVS already saturate around 80k steps, whereas CrossVS continues to improve and only reaches comparable performance of the others over 200k steps. Variant CrossVS shows a very slow convergence speed in speech generation task. The results in Appendix C.1 shows that, after 200 epochs of training, variant CrossVS still exhibits disadvantaged over the CrossV variant on both LRS2 and Chem dataset.
>
> ### 3. Experiment of scaling on data scope and model size (still ongoing)
>
> For data scope and model size, we compared the following two variants: CrossV and ConcatV. **They were trained using 1/2 of the original data volume and 1/2 of the original model size respectively.** These two ablation experiments are expected to be completed within 4 days, and we will immediately report the evaluation results.
>
> ### Citation
>
> [1] Peebles and Xie. Scalable Diffusion Models with Transformers. ICCV, 2023.
>
> [2]Chen et al. PixArt-α: Fast Training of Diffusion Transformer for Photorealistic Text-to-Image Synthesis. arXiv:2310.00426, 2023.
>
> [3] Esser et al. Scaling Rectified Flow Transformers for High-Resolution Image Synthesis. ICML, 2024.
>
> [4] Cheng et al. MMAudio: Taming Multimodal Joint Training for High-Quality Video-to-Audio Synthesis. CVPR, 2025.

---

> ### Author Response · Authors · 2025-11-24
> **Re3. Video Feature (In respond to Weakness.3)**
>
> ### 1. Video FPS does not work as the primary performance bottleneck for V2S.
>
> For example, recent V2A systems such as Diff-Foley and FRIEREN both operate successfully at only 4 FPS and still have high performance.
>
> ### 2. Experiments on Video features
>
> CLIP features possess strong semantic modeling capabilities and are therefore widely adopted in many Video-to-Speech (V2S) models. However, as pointed out by the reviewers, CLIP features provide less explicit temporal modeling than CAVP and Synchformer features. To directly address this concern, **we have launched a new training run that replaces CLIP features with CAVP features.** Training is expected to complete in ~2 days, and we will immediately update the evaluation result.
>
> ### Citation
> [1] Luo et al. Diff-Foley: Synchronized Video-to-Audio Synthesis with Latent Diffusion Models. NeurIPS, 2023.
>
> [2] Wang et al. Frieren: Efficient Video-to-Audio Generation with Rectified Flow Matching. NeurIPS, 2024.

---

### Author Response · Authors · 2025-11-24
**A Brief Summary for Our Rebuttal Response**

We sincerely thank all reviewers for their constructive and insightful feedback. We have improved the manuscript based on the comments received. Major revisions are highlighted in blue in the revised paper. Below is a concise summary of the key changes already incorporated:

1. For Reviewer Ssc7 and STq9: We provide a detailed description of the duration predictor's architecture and training procedure in the Appendix A. For the robustness of lip feature, we provide the evaluation results in our rebuttal response.

2. For Reviewer Ssc7: **We have trained a variant without any duration predictor, which shows better synchronization performance for VisualTTS task.** Details can be found in Appendix A.

3. For Reviewer Ssc7 and STq9: We **extended the training steps for all ablation variants**, and accordingly updated Figure 3 (a,b) and Appendix C.1.

4. For Reviewer STq9 and Xsf8: We **continue training the V2S-only and VisualTTS-only variants for another significantly 100k steps** and updated Figure.4 and the continued training results in Appendix C.2.

5. For Reviewer STq9: We added a new CFG ablation study on both the V2S and VisualTTS tasks in the Appendix D.

6. For Reviewer STq9 and Xsf8: We updated the VisualTTS benchmark results with E2TTS baseline and the V2S benchmark results with MMAudio baseline in Section 4.4.

For the remaining comments, some have been addressed in the rebuttal discussion. Below are additional experiments currently in progress. Results will be uploaded immediately upon completion:

1. For Reviewer Xsf8 and o5uM: We have randomly selected 20 samples from the V2S test set and 20 samples from the VisualTTS test set (40 samples in total) and submitted them to a professional third-party rating agency for **subjective evaluation**. Results are expected within 2–3 days and will be uploaded immediately once received.

2. For Reviewer Ssc7 and o5uM: We have launched a new training **replacing CLIP visual features with CAVP features**. Training is expected to finish in ~2 day, after which we will update all evaluation tables with the new results.

3. For Reviewer Ssc7 and o5uM: We have carried out experiments to **investigate the impact of model size and training data scope on different variants.** We compared the following two variants: CrossV and ConcatV. **They were trained using 1/2 of the original data scope and 1/2 of the original model size respectively.** These two ablation experiments are expected to be completed within ~4 days, and we will immediately report the evaluation results.

4. For Reviewer Stq9, to further investigate whether this overfitting alleviation is caused by the additional sound data itself, or by the model learning the conditional V2S task, we have launched a new controlled experiment: **training on VisualTTS plus unconditional V2S data (i.e., sound clips with their video conditioning deliberately blanked out, forcing the model to learn the unconditional distribution of sound).** This run is ongoing and expected to finish in ~5 days.

We believe these revisions and ongoing experiments can address the reviewers’ major concerns and further strengthen the paper. We look forward to further discussion and are happy to provide any additional clarification if needed.

---

### Meta-Review · Area_Chair_y6n8 · 2026-01-07

**Summary:**

This paper proposes VSSFlow, a unified flow-matching framework for video-to-sound (V2S) and VisualTTS. Reviewers’ main concerns centered on: (1) the unclear practical utility of video conditioning for TTS; (2) whether the architectural conclusions hold under scaling; (3) the clarity and necessity of key components (e.g., the duration predictor); (4) evaluation fairness/consistency and missing or insufficiently justified baselines; (5) the strength of evidence supporting the claimed mutual benefits of joint training; and (6) the lack of subjective evaluation and limited evidence using temporally precise video features (with CLIP at 10 FPS). During discussion, the authors added several ablations and clarifications (e.g., removing the duration predictor, extending training curves, CFG studies, adding baselines such as E2TTS and MMAudio, and explaining pipeline differences behind metric discrepancies), which resolved many issues around clarity and experimental completeness. However, at least one reviewer remained unconvinced about the interpretation of the joint-training benefits and the fairness of the evaluation protocol.

**Reviewer Concerns:**

***Concerns addressed by the rebuttal and discussion***

Duration predictor: The authors clarified the architecture and training procedure and added an explicit ablation removing the duration predictor.

Scaling with training steps.

Classifier-free guidance (CFG): Additional ablations on CFG scales were added for both V2S and VisualTTS tasks.

Missing baselines: The authors added E2TTS as a VisualTTS baseline and updated the V2S benchmarks to include MMAudio.

***Concerns still outstanding or only partially addressed***

Evaluation protocol fairness: Although the authors carefully identified preprocessing differences and re-evaluated results under multiple pipelines, Reviewer Xsf8 argued that this analysis highlights the sensitivity of the reported metrics to post-processing choices, raising doubts about why the chosen evaluation protocol should be considered fair or unbiased.

Joint training benefits: There remains disagreement on whether joint training provides meaningful benefits for VisualTTS beyond what could be achieved via early stopping of a VisualTTS-only model. In addition, the explanation of the observed CFG behavior contains a critical inconsistency and does not clearly establish the mechanism by which joint training improves performance.

Rationale for video conditioning in TTS: The rebuttal experiments show that performance is highly sensitive to the availability and quality of lip features. When video features are noisy or inconsistent with the speech content, the model may degrade significantly or fail. This validates Reviewer Ssc7’s concern and remains fundamentally unresolved.

Scaling beyond training steps: While extended training-step results were provided, ablations on data volume and model size were still ongoing at the time of the rebuttal and therefore do not yet support broader scaling conclusions.

Subjective evaluation: A subjective listening study was proposed but not yet reported in the discussion.

Temporal video features: Experiments replacing CLIP features with temporally aware features (e.g., CAVP) were proposed, but results were still pending.

The joint sound–speech generation part is interesting, but it currently lacks sufficient detail and rigorous evaluation. While the authors note practical constraints, these limitations prevent a thorough validation.

**Reviewer Scores:**

Reviewer Ssc7 (initial score: 4):
The authors added new experiments examining the necessity of the duration predictor, robustness to noisy lip features, and behavior under longer training schedules. While these additions improve clarity, they do not fully resolve the reviewer’s primary concern regarding the practical rationale for VisualTTS, particularly under realistic conditions where visual signals may be noisy or misaligned with speech.

Reviewer STq9 (initial score: 2):
The authors addressed several concrete issues, including extended training curves, step-aligned comparisons, CFG ablations, added TTS baselines, and additional implementation details. However, the reviewer’s strongest objections remain centered on the limited insight behind the proposed “inductive bias” explanation and the observation that adding visual modality can degrade TTS performance. Although the reviewer appreciated the joint sound–speech generation aspect, this component is presented only as a case study and does not constitute a fully developed or rigorously evaluated contribution in the current paper.

Reviewer Xsf8 (initial score: 4):
Despite the added experiments and clarifications regarding evaluation discrepancies, the reviewer’s follow-up comments indicate persistent concerns about evaluation protocol fairness, the strength of evidence supporting VisualTTS gains under joint training, and an apparent inconsistency in the CFG explanation.

Reviewer o5uM (initial score: 6):
The reviewer’s main requests were partially addressed through additional explanations and planned ablations, but not fully resolved with completed experiments.

Given that at least one reviewer remains unconvinced on the core claims, and that the evidence for “joint learning yields clear benefits” remains partially confounded by training dynamics and metric interpretation, the AC does not believe the paper is ready for acceptance in its current form.

---

### Decision · Program_Chairs · 2026-01-26

Reject